# $N^6$-methyladenosine RNA modification suppresses antiviral innate sensing pathways via reshaping double-stranded RNA

Weinan Qiu[1,6], Qingyang Zhang[2,3,6], Rui Zhang[4], Yangxu Lu[1], Xin Wang[1], Huabin Tian[1], Ying Yang[2,3], Zijuan Gu[1], Yanan Gao[1], Xin Yang[2,3], Guanshen Cui[2,3], Baofa Sun[2,3], Yanan Peng[1], Hongyu Deng[1], Hua Peng[1], Angang Yang[4], Yun-Gui Yang[2,3✉] & Pengyuan Yang[1,5✉]

Double-stranded RNA (dsRNA) is a virus-encoded signature capable of triggering intracellular Rig-like receptors (RLR) to activate antiviral signaling, but whether intercellular dsRNA structural reshaping mediated by the $N^6$-methyladenosine ($m^6A$) modification modulates this process remains largely unknown. Here, we show that, in response to infection by the RNA virus Vesicular Stomatitis Virus (VSV), the $m^6A$ methyltransferase METTL3 translocates into the cytoplasm to increase $m^6A$ modification on virus-derived transcripts and decrease viral dsRNA formation, thereby reducing virus-sensing efficacy by RLRs such as RIG-I and MDA5 and dampening antiviral immune signaling. Meanwhile, the genetic ablation of METTL3 in monocyte or hepatocyte causes enhanced type I IFN expression and accelerates VSV clearance. Our findings thus implicate METTL3-mediated $m^6A$ RNA modification on viral RNAs as a negative regulator for innate sensing pathways of dsRNA, and also hint METTL3 as a potential therapeutic target for the modulation of anti-viral immunity.

[1] Key Laboratory of Infection and Immunity of CAS, CAS Center for Excellence in Biomacromolecules, Institute of Biophysics, University of Chinese Academy of Sciences, Chinese Academy of Sciences, Beijing, China. [2] Key Laboratory of Genomic and Precision Medicine, Collaborative Innovation Center of Genetics and Development, China National Center for Bioinformation, Beijing Institute of Genomics, University of Chinese Academy of Sciences, Chinese Academy of Sciences, Beijing, China. [3] Institute of Stem Cell and Regeneration, Chinese Academy of Sciences, Beijing, China. [4] The State Key Laboratory of Cancer Biology, Department of Immunology, Fourth Military Medical University, Xi'an, Shaanxi, China. [5] Chongqing International Institute for Immunology, Chongqing, China. [6] These authors contributed equally: Weinan Qiu, Qingyang Zhang. ✉email: ygyang@big.ac.cn; pyyang@ibp.ac.cn

Pathogenic RNA viruses are considered as the primary etiological agents of human emerging pathogens and represent a challenge for global disease control. Therefore, it is critical to know the mechanism and interaction between RNA virus and host innate immunity. The innate immune system, highly conserved among plants and animals, recognizes the invading pathogens through pattern recognition receptors (PRRs) to trigger an effective immune response for defending the pathogens[1]. The virus-encoded molecular signatures, including cytosolic double-stranded RNA (dsRNA) and other distinct RNA species, trigger intracellular nucleic acid sensors, including retinoic acid-induced gene I (RIG-I) and melanoma differentiation-associated gene 5 (MDA5), to recognize these "non-self" RNAs and activate antiviral signaling[2–4]. Despite RIG-I and MDA5 show different preferences for sensing of 5′ ppp/ short dsRNA and long dsRNA, respectively, their recognition patterns and functional compensations for sensing foreign RNAs remain largely elusive[4]. On the other hand, viruses also evolve many mechanisms to weaken the host innate immune response through encoding immune response-neutralized proteins or modifications on viral nuclear acids to mimic host "self" components[5–7]. However, the precise mechanism of $N^6$-methyladenosine (m6A) modification for controlling innate sensing system remains largely unclear.

The RNA modification m6A is one of the most abundant mRNA modifications, which regulates several procedures of mRNA metabolism, especially the mRNA translation and degradation[8]. Accordingly, several m6A machines have been well identified, including m6A "writers" (e.g., METTL3, METTL14, and WTAP), "readers" (e.g., YTHDF1-3 and YTHDC1), and "erasers" (ALKBH5 and FTO)[9–11]. Based on the diverse functions of m6A machines, m6A modification has been shown to impact many fundamental biological processes including DNA damage repair[12], tumorigenesis[13–16], inflammation, and T-cell homeostasis[17]. In addition, m6A also functions in modulating the life cycle of various RNA and DNA viruses through m6A-mediated regulation of viral RNA processing[18,19]. Beyond that, recent studies demonstrated that m6A modification could regulate the innate immune responses via targeting mRNA stability of type I inteferons[20], host metabolic gene α-ketoglutarate dehydrogenase (OGDH)[21], or interfering the innate sensing of RIG-I through m6A "readers" proteins[5,22]. However, these reports mainly focused on the influence of host gene regulations for innate immunity. It remains obscure whether or how m6A modulates viral RNA structure or dsRNA loads to regulate the initial sensing pathways by host PRRs.

In this study, we identify METTL3 as a negative suppressor for global innate immune signaling cascades in response to infection of RNA virus Vesicular Stomatitis Virus (VSV) in vitro and in vivo. We find that METTL3 translocates to cytoplasm upon VSV infection and catalyzes the methylation of cytosolic viral positive-sense (+) RNAs. Subsequently, METTL3-mediated m6A modification reshapes viral RNA duplex structure and impairs viral RNA sensing efficacy by RIG-I and MDA5. These findings demonstrate the functional significance of METTL3-mediated m6A modification in innate sensing and provide a strong impetus for therapeutic intervention.

## Results

### METTL3 impedes global innate immune signaling cascades.
To fully investigate the potential roles of m6A modification in innate immunity, we screened the m6A machinery genes with overexpression of m6A "writers" and "erasers" in HeLa cells upon VSV infection, respectively. As shown in Supplementary Fig. 1a, the VSV-induced IFNB1 expression was significantly inhibited by m6A "writers", particularly the methyltransferase METTL3.

To further confirm the suppressive role of METTL3 in innate immunity, we treated cells with different triggers, including poly (dA:dT), poly(I:C), LPS, HBV, HCMV, HSV, HCV, SeV, and VSV, and observed a common inhibitory pattern of METTL3 on Ifnb1 expression (Fig. 1a and Supplementary Fig. 1b). Intriguingly, the Ifnb1 mRNA was most significantly decreased upon VSV infection within these triggers. Furthermore, we performed dual-luciferase reporter assay to test the role of METTL3 in the treatment-induced activation of IFN regulatory factor 3 (IRF3), the key upstream transcription factor of type I IFNs (Fig. 1b and Supplementary Fig. 1c). We found that the IFN-β promoter activity was significantly inhibited by overexpressed METTL3 in a dose-dependent manner upon VSV infection (Fig. 1b and Supplementary Fig. 1c), suggesting that METTL3 regulates the transcription of Ifnb1 through the upstream of innate immune signaling when responding to VSV infection. Consistently, the deficiency of METTL3 repressed VSV transcripts and titer, meanwhile increasing the expression of IFN-β in RAW264.7 cells. (Supplementary Fig. 1d–j). Together, these data suggest that METTL3 promotes VSV immune escape through attenuating type 1 IFN signaling activation.

We next focused on the regulatory mechanism of METTL3 in VSV-induced innate immune pathway. Given that the global depletion of METTL3 leads to embryonic death[23], we crossed the Mettl3$^{flox/flox}$ mice with Lyz2-Cre mice to generate monocyte-specific Mettl3-deficient mice (termed Mettl3$^{f/f}$ Lyz2-Cre hereafter) (Supplementary Fig. 1k). Consistent with previous reports, we also observed an enhanced downstream of innate immunity[20], phosphorylation of ISGs-regulated protein Stat1, in the Mettl3-deficient macrophages upon VSV infection, which was supposed to be increased by a decrease of m6A-modified Ifnb1 mRNA. However, more importantly, we observed that the phosphorylation of upstream kinase Tbk1, transcription factors p65 and Irf3, was also significantly increased in Mettl3-deficient macrophages (Fig. 1c), indicating a global influence for innate immune signaling cascades by Mettl3. To confirm the observed inhibition on activation of IRF3, we transiently knocked down or overexpressed METTL3 in different cell lines upon VSV infection. Consistently, transient silence of METTL3 promoted the phosphorylation of IRF3 in A549 cells (Fig. 1d), while overexpression of METTL3 in RAW264.7, HeLa, Huh7, and LO2 cells suppressed the activation of IRF3 (Fig. 1e–h). Obviously, immunofluorescent results indicated that the nuclear translocation of IRF3 was almost inhibited by METTL3 upon VSV infection, in contrast to the robust phosphorylated IRF3 observed in the negative control cells (Fig. 1i and Supplementary Fig. 1l). These findings reveal that METTL3 is a strong intrinsic inhibitor for repressing the upstream of innate immune signaling in various cells.

To evaluate the transcriptome-wide role of METTL3 in VSV-induced signaling pathways, we isolated the total RNA from WT and Mettl3-depleted RAW264.7 cells exposed to PBS or VSV treatment, and carried out RNA-sequencing (RNA-seq). Both GO enrichment and Reactome pathway analysis suggested that upregulated genes are mainly involved in the innate immune pathway after METTL3 depletion (Fig. 1j and Supplementary Fig. 1m). Gene set enrichment analysis (GSEA) reflected the activation of genes involved in innate immune response after METTL3 depletion (Fig. 1k). Moreover, in Mettl3-depleted RAW264.7 cells, 1160 upregulated and 1317 downregulated genes were detected after VSV infection, respectively (Fig. 1l and Supplementary Fig. 2a); and among which, most of IFNs-stimulated genes (ISGs) were upregulated (Fig. 1l and Supplementary Fig. 2a, b). By contrast, in PBS mock infection, only 299 upregulated and 199 downregulated genes were detected and most of ISGs expression were not changed in Mettl3-depleted

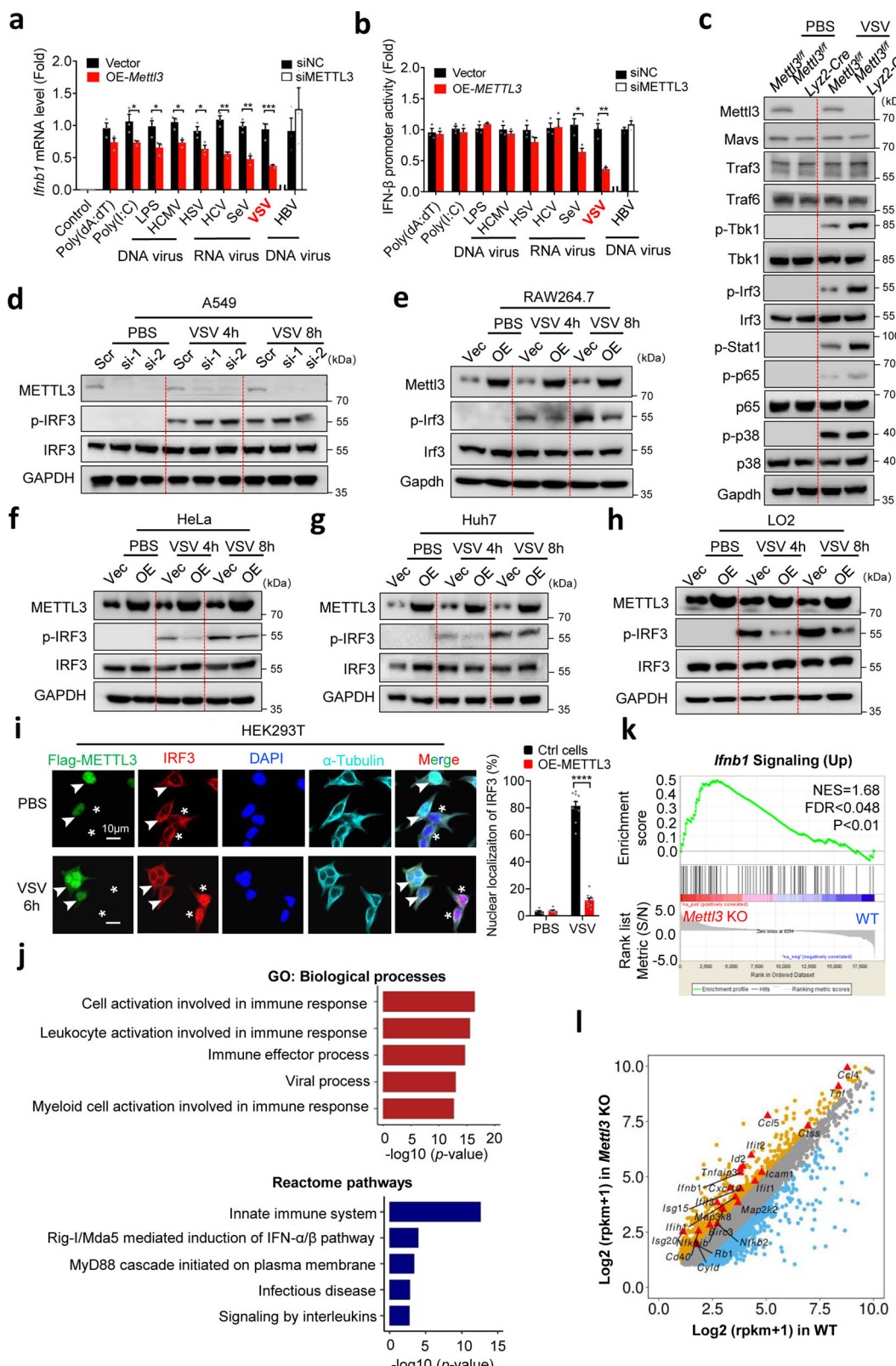

RAW264.7 cells (Supplementary Fig. 2a–c). Overall, these results demonstrate that METTL3 could globally interfere with innate immune signaling upon VSV infection.

**Targeting METTL3 enhances innate immune response and viral clearance in vivo.** To fully address the crucial role of METTL3 in innate immunity in vivo, we intravenously infected *Mettl3*^f/f *Lyz2-Cre* mice and the littermate control *Mettl3*^f/f mice with VSV. In line with the in vitro results, the *Mettl3*^f/f *Lyz2-Cre* mice were significantly resistant to high dose of VSV-induced lethality compared with control mice (Fig. 2a). When we challenged the mice with a moderate dose of VSV, *Mettl3*^f/f *Lyz2-Cre* mice displayed less of the VSV-induced pathologic lesions in the lung and liver at 24 h after infection than control mice (Fig. 2b, c

**Fig. 1 METTL3 globally inhibits innate immune signaling cascade. a** qPCR analysis of *Ifnb1* expression following a 12 h treatment with different triggers as indicated before overexpression of Mettl3 in RAW264.7 cells (Left). qPCR analysis of *IFNB1* expression following a 12 h treatment with HBV before knockdown of METTL3 in AC12 cells (Right). n = 3 biologically independent experiments. **b** IFN-β promoter activity in HEK293T cells transfected with METTL3 vector upon different treatments for 12 h (Left). IFN-β promoter activity in AC12 cells transfected with siMETTL3 upon HBV treatment for 12 h (Right). n = 3 biologically independent experiments. **c** Western blot analysis of the proteins in innate immune signaling isolated from peritoneal macrophage of WT or *Mettl3*-deficient mice. The peritoneal macrophage was treated with PBS or VSV infection for 12 h before lysis. **d** Western blot analysis of the proteins from A549 cell line before treated with siScramble or siMETTL3 for 2 days. **e–h** Western blot analysis of the proteins from RAW264.7 (**e**), HeLa (**f**), Huh7 (**g**), and LO2 (**h**) cell lines before overexpression of METTL3 for 2 days. **i** Immunofluorescence analysis of IRF3 translocation and activation in HEK293T cells after VSV infection for 6 h or PBS treatment. The arrowed cells indicated the METTL3 overexpressed cells which inhibited the nuclear translocation and activation of IRF3. The "*" labeled cells indicated control cells. α-Tubulin indicated the cytosolic part. n = 2 biologically independent experiments. **j** GO and pathway enrichment analysis of upregulated genes upon Mettl3 knockout in RAW264.7 cells after VSV infection (two-tailed hypergeometric test with $p < 0.05$). **k** Two-tailed Gene Set Enrichment Analysis of the *Ifnb1* signaling transcriptional signature in Mettl3 knockout relative to control groups. NES, normalized enrichment score. **l** Scatter plot showing the alteration of genes expression by comparing Mettl3 knockout and WT RAW264.7 cells after VSV infection for 12 h (yellow, upregulated; blue, downregulated; gray, no significant change; red triangle, ISGs). Data are representative of 2–3 independent experiments. $*p < 0.05$, $**p < 0.01$, $***p < 0.001$, as determined by two-tailed unpaired Student's $t$ test (Fig. 1a, b, i). Error bars represent mean ± SEM.

and Supplementary Fig. 3a). Consistently, the qRT-PCR and Western blot analysis showed less viral replication, but higher *Ifnb1* mRNA in the spleen, liver, and lung of *Mettl3*<sup>f/f</sup> *Lyz2-Cre* mice compared to the control mice (Fig. 2d, e). Accordingly, the IFN-β secretion was significantly increased in the serum of VSV-infected *Mettl3*<sup>f/f</sup> *Lyz2-Cre* mice compared to the controls (Fig. 2f). To examine the in vivo upregulated IFN-β was dependent on enhanced upstream signaling of innate immunity, we collected peritoneal macrophages from infected mice. Consistent with the in vitro results in Fig. 1, Western blot indicated that METTL3 deficiency also promoted phosphorylated Irf3 in vivo (Fig. 2g). These results demonstrate that METTL3 deficiency enhances innate immune response for viral clearance in vivo.

On consideration of other types of cells such as hepatocytes also express PRRs to defend pathogens[24], we crossed the *Mettl3*<sup>f/f</sup> mice with *Alb-Cre* mice to generate hepatocyte-specific *Mettl3*-deficient mice (called it *Mettl3*<sup>f/f</sup> *Alb-Cre* mice hereafter), and confirmed the specific depletion of METTL3 in liver cells through Western blot, qRT-PCR, and dot blot analysis (Fig. 2h and Supplementary Fig. 3b, c). Consistent with *Mettl3*<sup>f/f</sup> *Lyz2-Cre* mice, *Mettl3*<sup>f/f</sup> *Alb-Cre* mice decreased VSV-induced lethality compared with control mice after a high dose of VSV infection (Fig. 2i). Upon a moderate dose of VSV, *Mettl3*<sup>f/f</sup> *Alb-Cre* mice also displayed enhanced antivirus ability through upregulating the expression of *Ifnb1* and ISGs in lung and liver compared with controls (Fig. 2j, k and Supplementary Fig. 3d, e). Consistently, the secretion of IFN-β was elevated in serum of *Mettl3*<sup>f/f</sup> *Alb-Cre* mice compared with controls post 24 h of VSV infection (Fig. 2l). Taken together, these two kinds of conditional gene editing mice demonstrated that METTL3 acts as a negative regulator in RNA virus-triggered innate immune response in vivo, which suggests METTL3 could be a potential target for antiviral therapy.

**VSV infection induces METTL3 cytoplasmic translocation and dampens type I IFNs.** Based on our observed global interference of METTL3 for innate immune signaling in contrast to current model that m⁶A-mediated destabilization of *IFNB1* mRNA by nuclear METTL3, we hypothesized that viral infection may influence METTL3 expression or subcellular pattern to further regulate upstream of innate sensor signaling cascades. Upon VSV infection, neither Mettl3 mRNA nor protein level was altered in RAW264.7 cells (Fig. 3a and Supplementary Fig. 4a). Interestingly, we found that VSV and SeV infections, but not other treatments including HBV, HSV, and poly(dA:dT) and poly(I:C), robustly enhanced cytoplasmic translocation of METTL3 from nucleus (Fig. 3b–d and Supplementary Fig. 4b), while the nuclear membranes were intact determined by Lamin A/C staining (Fig. 3c), demonstrating that

the translocation of METTL3 in the cell was a natural phenomenon in response to infection rather than resulted from the breach of the nuclear membrane. As some studies demonstrated that m⁶A modification is mediated by METTL3–METTL14 complex[25], some investigations reported that these two proteins could function independently[13,26]. We found that VSV infection cannot regulate the translocation of METTL14 (Supplementary Fig. 4c). Next, we asked whether this cytoplasmic translocation influences METTL3 catalytic activity. To address this question, we mutated the nuclear localization sequence (NLS) of METTL3 (Fig. 3e) and confirmed its cytoplasmic localization by immunofluorescence (Fig. 3f). The dot blot analysis revealed that NLS-mutated METTL3 displayed similar methyltransferase activity with wild-type (WT) METTL3 (Fig. 3g). Furthermore, when we overexpressed WT-METTL3 or NLS-mutated METTL3 in HeLa cells, NLS-mutated METTL3 even more significantly suppressed *IFNB1* expression upon VSV infection (Fig. 3h). And the further luciferase assay showed that overexpression of NLS-mutated METTL3 was sufficient to inhibit the upstream of innate immunity upon VSV infection (Fig. 3i). These data demonstrate that VSV induces METTL3 cytoplasmic localization and inhibits type I IFNs activation.

To examine whether the inhibitory effect of METTL3 on innate immunity is dependent on its m⁶A catalytic activity, we mutated the catalytic domain of METTL3 and confirmed its loss of methyltransferase function by dot blot (Supplementary Fig. 4d, e). The gain-of-function results demonstrated that METTL3-mediated m⁶A enhanced VSV titer and viral mRNA (Fig. 3j and Supplementary Fig. 4f). Consistently, the suppressed IFN-β expression and secretion by overexpression of METTL3 were eliminated by catalytic domain mutation (Fig. 3k, l). Taken together, VSV induces METTL3 cytoplasmic translocation to suppress type I IFNs and promote viral immune escape through its methyltransferase function.

**METTL3 mediates m⁶A modification on viral positive-sense RNAs.** Due to the inhibitory effect of METTL3-mediated methylation in multiple upstream cascades of innate immune signaling, we hypothesized that METTL3-mediated m⁶A modification on viral RNA may modulate the initial innate sensing pathways. To test whether METTL3 mediated viral RNAs methylation, we purified extracellular VSV to extract pure viral RNAs, incubated in vitro with Flag-METTL3 purified by immunoprecipitation (IP) from overexpressed HEK293T cell lysis (Fig. 4a and Supplementary Fig. 5a). The RNA dot blot result showed that the m⁶A level of VSV RNA was relatively low but was significantly enriched after incubation with purified Flag-METTL3 (Fig. 4b). We next employed the anti-m⁶A IP-qPCR (MeRIP-qPCR) to analyze the m⁶A level and

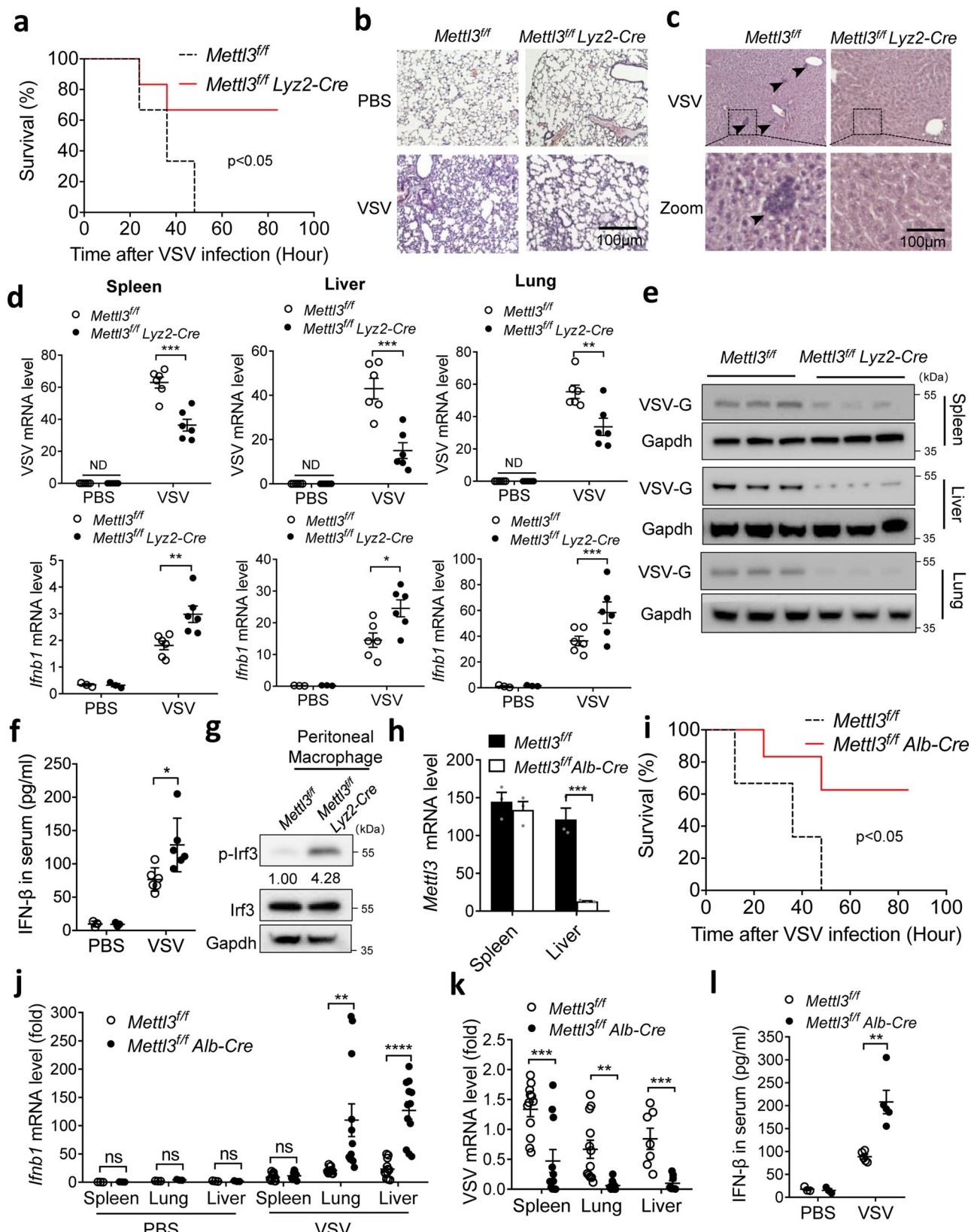

confirmed the enhanced VSV RNA m6A level upon METTL3 overexpression (Fig. 4c). Consistently, RNA-FISH result suggested the co-localization between VSV RNAs and m6A modification in the cytoplasm (Fig. 4d). To further confirm the RNA-binding regions of METTL3 and m6A sites, we carried out photoactivatable-ribonucleoside-enhanced crosslinking and immunoprecipitation

(PAR-CLIP) sequencing and miCLIP-seq. Consistent with a recent report[20], METTL3 and m6A showed specific binding regions on *Ifnb1* transcript (Supplementary Fig. 5b–e). Moreover, as VSV RNAs include the positive-sense (+) RNAs and negative-sense (−) genomic RNA, through strand-specific sequencing and analyzing, we found that only VSV (+) RNAs showed abundant METTL3

**Fig. 2 Deficiency in METTL3 protects mice against VSV infection in vivo. a** Survival of *Mettl3*[f/f] *Lyz2*-Cre and *Mettl3*[f/f] littermate female mice (*n* = 5 per group) at various times (horizontal axes) after intraperitoneal infection with VSV (1×10$^9$ PFU per mouse). **b, c** Microscopy of hematoxylin-and-eosin-stained lung (**b**) or liver (**c**) sections from female mice treated with PBS or VSV. **d** qRT-PCR analysis of *Ifnb1* and VSV mRNA in the spleens (left), livers (middle), and lung (right) of *Mettl3*[f/f] *Lyz2*-Cre and *Mettl3*[f/f] littermate female mice (*n* = 5 per group) given intraperitoneal injection of PBS or infected for 24 h by intraperitoneal injection of VSV (2.5×10$^8$ PFU per mouse); results are presented relative to those of *actin*. **e** Western blot analysis of VSV-G in the spleens, livers, and lungs of infected mice. **f** ELISA analysis of IFN-β in serum. *n* = 6 biologically independent animals. **g** Isolated peritoneal macrophages from VSV-infected mice, and then performed Western blot by indicated antibodies. **h** qRT-PCR analysis of Mettl3 conditionally knocked out in hepatocytes. *n* = 3 biologically independent experiments. **i** Survival of *Mettl3*[f/f]*Alb*-Cre and *Mettl3*[f/f] littermate female mice (*n* = 5 per group) at various times (horizontal axes) after intraperitoneal infection with VSV (1×10$^9$ PFU per mouse). **j, k** qRT-PCR analysis of *Ifnb1* (**i**) and VSV (**j**) mRNA in the spleens, livers, and lung of *Mettl3*[f/f] *Alb*-Cre and *Mettl3*[f/f] littermate female mice (*n* = 5 per group) given intraperitoneal injection of PBS or infected for 24 h by intraperitoneal injection of VSV (2.5×10$^8$ PFU per mouse); results are presented relative to those of *actin*. **l** ELISA analysis of IFN-β in serum. *n* = 5 biologically independent animals. *$p$ < 0.05, **$p$ < 0.01, ***$p$ < 0.001, ****$p$ < 0.0001 as determined by two-tailed unpaired Student's *t* test (**d, f, h, j–l**). Log-rank (Mantel–Cox) test (**a, i**). Error bars represent mean ± SEM.

binding sites (Fig. 4f), which strongly supported our hypothesis that VSV infection triggers METTL3 cytosolic translocation to interact with VSV RNAs. In addition to the METTL3 PAR-CLIP-seq results, miCLIP-seq also identified the accurate m$^6$A modification sites on VSV (+) RNAs while not VSV (−) RNA (Fig. 4e, f). Importantly, the m$^6$A and METTL3 binding sites are coincidently accumulated at 3′ and 5′ terminals of N, P, M, G viral (+) RNAs (Fig. 4f). To better quantify the m$^6$A abundance of these viral RNA methylated sites in WT and *Mettl3*-deficient RAW264.7 cells, we performed MeRIP-qPCR assay and identified 18 m$^6$A positions on viral (+) RNAs that were significantly reduced in *Mettl3*-deficient cells. (Fig. 4g and Supplementary Fig. 5f). Together, these results demonstrate that METTL3 binds to VSV (+) RNAs and mediates m$^6$A modification.

**METTL3-mediated m$^6$A modification reshapes viral dsRNA structure.** Since recent studies reported that m$^6$A "reader" proteins suppress innate immunity sensing via compete with RLRs to foreign RNA[5,22], indeed, we found that YTHDF2 could bind to VSV (+) RNA (Supplementary Fig. 6a), but knockdown of YTHDF2 did not entirely rescue the expression of *IFNB1* upon transfection with m$^6$A-modified VSV (+) RNA oligo (biotin-m$^6$A2,964) (Supplementary Fig. 6b, c). This suggests that it remains alternative important mechanisms behind the phenotype.

To date, it is widely accepted that virus-encoded dsRNA triggers cellular sensors to recognize as foreign distinct RNA signature to launch the innate immune responses[27,28]. However, it remains unclear whether and how viral dsRNA is modified by host cells. Consistent with previous studies, we also observed abundant dsRNA foci in cytoplasm induced by VSV infection (Fig. 5a). Furthermore, we analyzed the role of METTL3 in dsRNA formation upon VSV infection. Interestingly, dsRNA loads were significantly increased in the METTL3 knockdown cells compared to the control cells (Fig. 5a, b). To investigate the role of endogenous METTL3-mediated RNA modification on VSV infection-derived dsRNA, we purified dsRNA by anti-dsRNA (J2)-immunoprecipitation (dsRIP) with confirming the efficiency of the anti-J2 antibody for dsRIP and measured their m$^6$A level by dot blot (Fig. 5c). Importantly, we observed a reduced m$^6$A level in immunoprecipitated dsRNA in METTL3 knockdown HeLa cells (Fig. 5d), revealing that METTL3 influenced dsRNA formation after VSV infection. Based on our identification of the m$^6$A-modified positions of VSV RNAs (Fig. 4f, g), we speculated that m$^6$A modification may influence the formation of dsRNA on these sites. As shown in Fig. 5e, dsRIP-qPCR results depicted an increase of dsRNA level among majority of these identified m$^6$A positions on VSV RNA upon METTL3 ablation (Fig. 5e). Overall, these data suggest that METTL3-mediated m$^6$A modification on VSV RNA impairs the formation of dsRNA.

**m$^6$A modification impairs viral RNA sensing efficacy by RLRs.** Based on our observation that m$^6$A modification impairs the formation of dsRNA, we next asked whether METTL3-mediated m$^6$A could influence the recognition of RLRs, including Mda5 and Rig-I, for viral RNAs. To identify their RNA binding sites, we firstly performed PAR-CLIP-seq analysis for Mda5 and Rig-I, accompanied with miCLIP-seq to display the m$^6$A peak regions. We found that both Mda5 and Rig-I were less prone to bind on VSV (−) genomic RNA (Supplementary Fig. 7a, b), and showed no peaks on *Gapdh* mRNA as negative control (Supplementary Fig. 7c). As shown in Fig. 6a, b, we identified abundant binding clusters of Mda5 and Rig-I on VSV (+) RNAs, which displayed high correlations with the m$^6$A regions (Fig. 6a, b). However, hyper-methylation level of m$^6$A positions was matched with poor binding regions of Mda5 or Rig-I, and vice versa, suggesting that m$^6$A modification negatively controls the binding of RLRs to VSV dsRNA. To exclude the impact of METTL3-mediated m$^6$A to RLR expressions, we performed Western blot and observed METTL3 depletion did not influence the basal level of RLRs, including Rig-I and Mda5 (Supplementary Fig. 7d). Consistent with our data in Fig. 1, METTL3 impedes global innate cascades, these data also suggest the potential role of m$^6$A modification in innate sensing efficacy.

To investigate the impact of m$^6$A modification on RNA binding efficacy of RLRs, we synthesized two biotin-labeled viral RNA sequences for RNA pulldown assays at the m$^6$A-correlated RLRs peak regions (from VSV (+) RNA 2912–2974 nt) that, respectively, contained variants including an unmodified nucleotide (biotin-A2,964) and a m$^6$A modified mimic nucleotide (biotin-m$^6$A2,964) (Supplementary Fig. 7e). Significantly, the biotin-m$^6$A2,964 bound weakly to Rig-I and Mda5 proteins compared with the biotin-A2,964 (Fig. 6c). To demonstrate that the m$^6$A modification disrupts viral dsRNA formation and changes the affinity of RLRs on m$^6$A-modified dsRNA, we tested the binding affinity of biotin-A2,964/ biotin-m$^6$A2,964 to J2 antibody by immunoprecipitation. We found m$^6$A modification indeed decreased the binding between J2 antibody and synthesized VSV RNA oligo (Supplementary Fig. 7f), suggesting that m$^6$A modification impairs dsRNA formation on VSV RNA to attenuate RLRs sensing. Additionally, we synthesized biotin-poly(A:U), the widely used mimic for dsRNA, together with the methylated biotin-poly(m$^6$A:U) to pulldown RLRs (Supplementary Fig. 7g), and observed that m$^6$A modification attenuated the interaction between poly(A:U) and RLRs (Supplementary Fig. 7h). Consistently, methylated poly(m$^6$A:U) failed to induce TBK1-IRF3-IFNb1-ISGs signaling activation in sharp contrast to the robust induction by poly(A:U) (Supplementary Fig. 8a–f). These data demonstrate that m$^6$A interferes with the recognition of RLRs on Viral RNAs and abolishes the RLR-mediated innate sensing pathways. To understand the endogenous role of

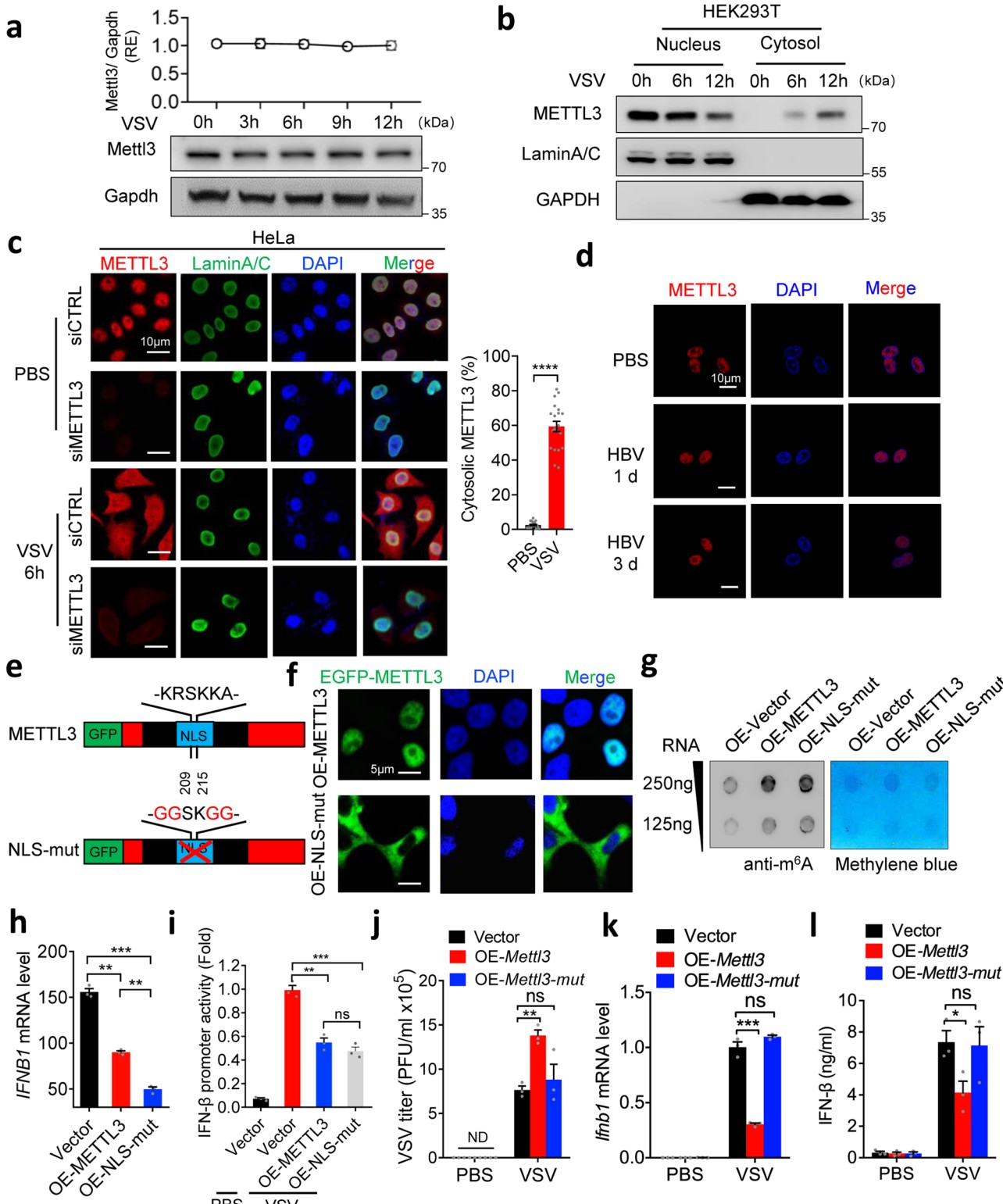

METTL3, we knocked down METTL3 in HeLa cells to determine the influence on intercellular innate sensing efficacy of RLRs. Compared to the control, METTL3-depleted cells displayed more co-localization of RLRs and dsRNA intracellularly upon VSV infection (Fig. 6d). Consistently, METTL3 depletion significantly promoted the interaction of RLRs with VSV RNAs by RIP-qPCR (Fig. 6e). Taken together, these results strongly support our hypothesis that METTL3-mediated m6A modification plays as an innate immune suppressor to inhibit

the sensing of RLRs to viral RNAs directly, which suggests that m6A-modified viral RNA motif acts as cis-acting element to control innate sensing through decreasing dsRNA formation and dsRNA loads (Fig. 7).

## Discussion
In this study, we extensively uncover a signature network that METTL3-mediated m6A RNA modification on viral RNA as a

**Fig. 3 VSV infection induces METTL3 cytoplasmic translocation and dampens type I IFNs. a** Western blot analysis of Mettl3 protein along VSV infection of RAW264.7 cells. The upper graph indicates statistic result. **b** Cellular fractionation and Western blot analysis of METTL3 protein localization before or post VSV infection in HEK293T cells. **c** Immunofluorescence analysis of endogenous METTL3 protein subcellular localization before or post VSV infection for 6 h in HeLa cells. **d** Immunofluorescence analysis of endogenous METTL3 protein subcellular localization before or post HBV (DNA virus) infection as indicated timepoints in AC12 cells. **e** The domain organization and dissection of the human METTL3 protein. NLS-mut: nuclear localization signal-mutation. **f** Immunofluorescence analysis of METTL3 protein subcellular localization after NLS-mutation in HeLa cells. **g** Dot blot analysis of mRNA-m6A level after overexpression of NLS-mut or WT METTL3 in HeLa cells. Methylene blue staining indicated the loading control. **h** qRT-PCR analysis of *IFNB1* expression following a 12 h treatment with PBS or VSV infection before transfection of Vector/ NLS-mut/ WT METTL3, respectively, in HeLa cells. **i** IFN-β promoter activity in HEK293T cells transfected with METTL3 vectors upon VSV infection. **j** RAW264.7 cells overexpressed catalytic-mutated Mettl3 (Mettl3-mut) or WT Mettl3 in mock (PBS) or VSV infection for 12 h. The supernatants were collected to perform plaque assay, which indicated the VSV titer. **k, l** qRT-PCR and ELISA analysis of *Ifnb1* expression in RAW264.7 cells overexpressed Mettl3-mut or WT Mettl3 in mock (PBS) or VSV infection for 12 h. Data are representative of 3 independent experiments. *$p < 0.05$, **$p < 0.01$, ***$p < 0.001$, as determined by two-tailed unpaired Student's *t* test. Error bars represent mean ± SEM.

"self" marker to suppress innate sensing pathways through decreasing viral dsRNA formation. Beyond the recent reports that m6A modification controls innate immune response via targeting mRNA stability of host type I IFN or OGDH[20,21], our study revealed that m6A methylation significantly inhibits upstream signaling of innate immunity in various cell lines. Moreover, METTL3 can be translocated from the nucleus to the cytosol upon VSV infection to directly methylate viral RNAs. Consequently, the m6A modified viral RNAs were reshaped with reduced dsRNA loads to suppress innate sensing efficacy by MDA5 or RIG-I, which inhibits global innate immune signaling pathways. In a murine model, METTL3 depletion in monocyte or hepatocyte protects the mice against VSV infection and enhances type I IFN greatly in vivo, presenting the potential therapeutic applications for viral infectious therapy.

The immune response in human beings undergoes a "butterfly effect", initiated from intracellular innate immune sensing to a global adaptive immune response[29]. Virus-encoded molecular signatures, such as cytosolic dsRNA or otherwise foreign "non-self" RNA species, trigger cellular antiviral innate immune signaling, which is beneficial for the maintenance of body homeostasis during infections[29–32]. It is widely accepted that cytosolic RNA sensors RLRs, RIG-I, and MDA5 recognize these "non-self" RNAs and activate signal transduction pathways to induce a status of antiviral defense with expression of type I IFNs[27]. However, the PRRs-triggered elevation of these antiviral cytokines may also lead to severe adverse effects with autoimmune responses[33,34]. How the host cells modulate the immune sensor triggered signaling activation within a scope of control remains largely obscure. In a dsRNA-dependent manner, our present findings reveal a new negative regulation pathway that METTL3 is translocated to the cytoplasm upon VSV infection to methylate viral RNA, and orchestrates innate signaling homeostasis via m6A modification to decrease dsRNA level. From the virus side, our study also elucidates a strategy for virus immune escape that virus hijacks m6A modification to disrupt dsRNA formation to escape RLR sensing.

Although we identified METTL3 as an innate immune rheostat for a broad spectrum of pathogen triggers, the data suggests that METTL3 is more efficient for RNA virus-induced innate immunity, especially for VSV, a negative-sense RNA virus that widely used for innate immune studies. In contrast to DNA virus and some RNA virus-like Influenza virus, VSV RNAs, including genomic (−) RNA and (+) RNAs, mainly localize in the cytoplasm. Based on our finding that METTL3 globally suppresses innate upstream pathways, we further identified that VSV infection attracts METTL3 from the nucleus to the cytoplasm. Intriguingly, this cytosolic translocation did not influence its methyltransferase activity, but cytosolic METTL3 significantly inhibited interferon expression which depended on its catalytic activity. Although our

findings reveal that VSV could recruit cytosolic METTL3 to suppress innate immunity, the intrinsic mechanism on how METTL3 translocated from the nucleus to cytosol upon VSV infection needs further investigation.

Emerging evidence indicates that m6A modification influences various RNA metabolisms, which is largely dependent on different m6A "reader" proteins[8,10]. In many cases, YTHDF1, YTHDF3, and YTHDC2 promote the translation of m6A-methylated mRNA, while YTHDF2 enhances translation and decay[9–11]. For the innate immunity sensing, m6A "reader" proteins can function as trans-acting factors to compete with and suppress RLRs to bind to m6A-modified RNAs[5,22]. However, whether and how m6A-modified RNA motif plays as a *cis*-acting element to regulate innate sensing remains unclear. On the other hand, m6A modification also displays crucial roles in reshaping the intrinsic RNA secondary structure formation[35,36]. However, whether foreign viral RNA undergoes a similar mechanism remains largely unknown. Our findings identified the significant increase of dsRNA level in VSV-treated METTL3-depleted cells, which specifically promoted dsRNA formation at the m6A-methylated sites of viral RNA (Fig. 5b, e). The PAR-CLIP-seq data demonstrate that both RIG-I and MDA5 bound to VSV (+) RNAs directly, particularly at the m6A modified regions. Previous studies reported that RIG-I and MDA5 show different recognition preferences for different dsRNA species[4,22]. In our study, we demonstrated that RIG-I and MDA5 have highly correlated recognition patterns on VSV (+) RNAs, suggesting the direct sensing and functional compensation between RIG-I and MDA5 to viral single-stranded RNA-folded dsRNA. Moreover, m6A-modified dsRNA poly(m6A:U) and VSV (+) RNA oligo indeed disrupted the recognition and activation of RLRs (Fig. 6c and Supplementary Fig. 7d, e). Thus, our findings highlight a ubiquitously biological phenomenon that viruses and host cells could orchestrate the m6A modification for foreign RNAs to mimic "self" component by reshaping the dsRNA structures.

A recent study demonstrated that m6A modification did not alter the phosphorylation of IRF3 by DNA virus HCMV infection[37]. In consideration of the differences between DNA virus and RNA virus (Fig. 1a, b), we detected the TBK1-IRF3 signaling in various cell lines after VSV infection. Intriguingly, METTL3 inhibited TBK1-IRF3 activation under VSV treatment. Our explanation is that RNA virus containing more species of RNAs, including genomic, replicative intermediate RNA and transcript RNAs, might be regulated by m6A and sensed by RLRs to induce innate immunity. Notably, a recent paper reported that RNA m6A modification attenuates the sensing of RIG-I dependent on m6A "reader" proteins[22], but how these "reader" proteins function for RIG-I sensing remains uncovered. It is worth noting that our miCLIP and PAR-CLIP-seq data revealed that METTL3 mediated m6A modification on *Ifnb1*, consistent with the reported study[20,37]. Hence, METTL3 seems to globally suppress

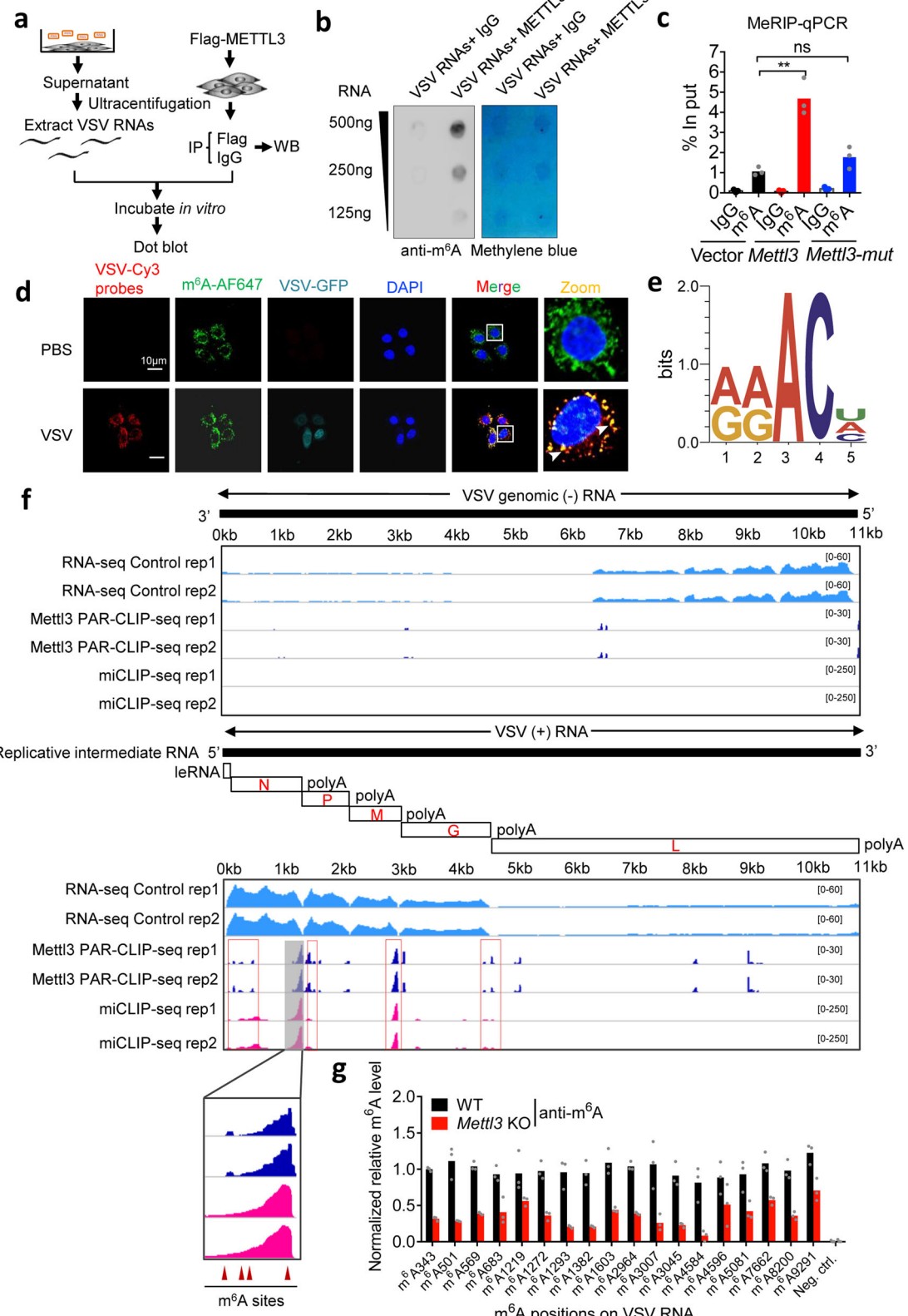

**Fig. 4 METTL3 mediates m⁶A modification on VSV RNA. a** A schematic representation of the experimental procedure used in **b**. **b** Dot blot analysis of VSV RNA m⁶A level treated with immunoprecipitated (IP)-IgG or IP-METTL3, respectively. Methylene blue staining indicated the loading control. **c** MeRIP-qPCR analysis of VSV RNA m⁶A level following treatment with overexpression of WT or Mettl3-mut, respectively. $n = 2$ biologically independent experiments. **d** RNA-FISH analysis of the co-localization of VSV RNA and m⁶A modification in the cytosol. **e** The conserved sequence motif of m⁶A residues in CIMS-based miCLIP-seq. **f** Integrative genomics viewer (IGV) plots of the METTL3-binding regions and m⁶A modification on VSV negative sense (−) genomic RNA (Upper graph) and VSV positive sense (+) RNAs (Lower graph). m⁶A sites are indicated by red triangles. **g** MeRIP-qPCR analysis of specific m⁶A sites on VSV RNA in WT or Mettl3 KO RAW264.7 cells. $n = 2$ biologically independent experiments. *$p < 0.05$, **$p < 0.01$, as determined by two-tailed unpaired Student's $t$ test (**c**, **g**). Error bars represent mean ± SEM.

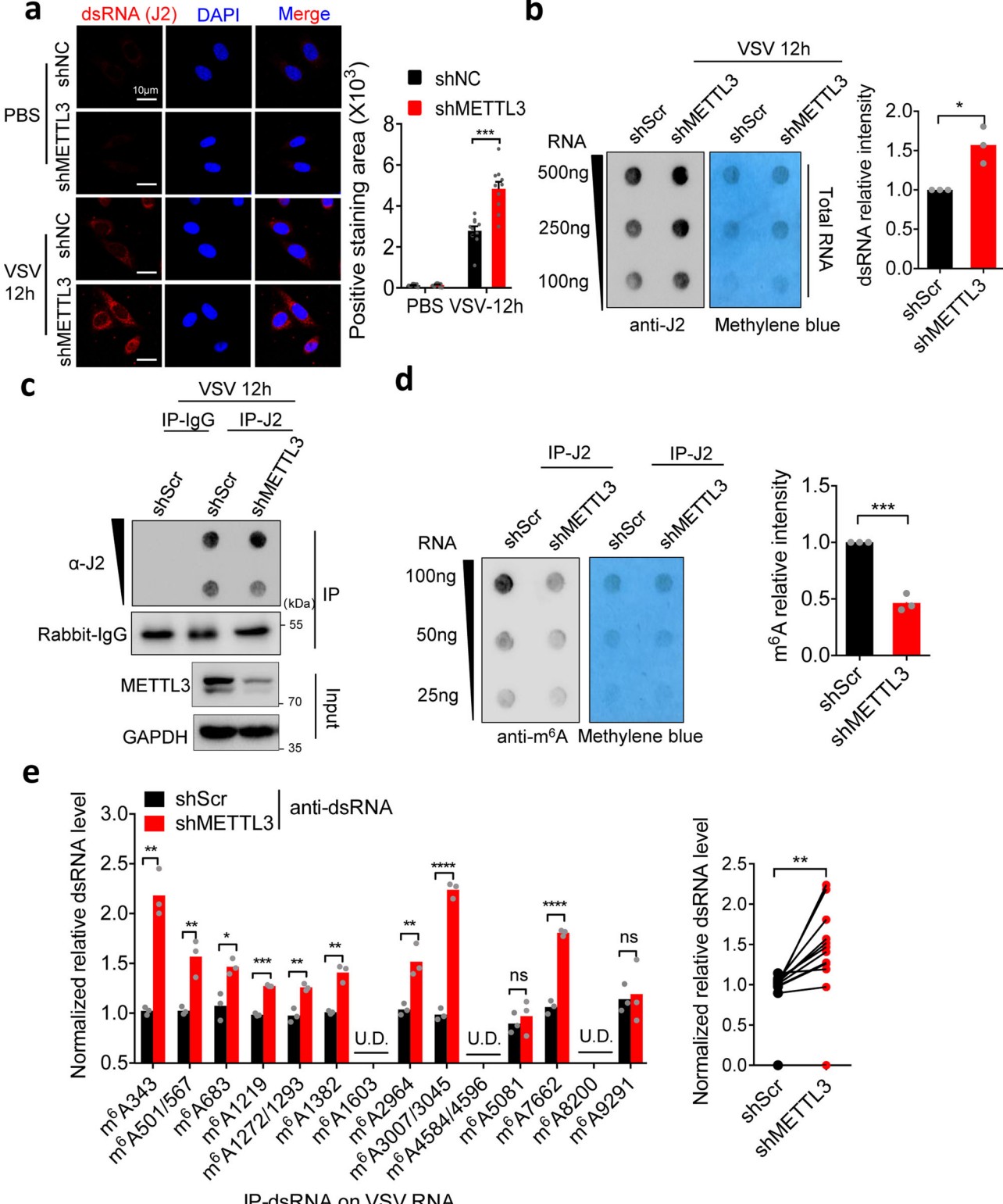

**Fig. 5 METTL3-mediated m⁶A modification reshapes viral dsRNA.** **a** Immunofluorescent analysis of dsRNA level after PBS treatment or VSV infection for 12 h in shNC and shMETTL3 HeLa cells. $n = 10$ cells examined over 2 independent experiments. **b** Dot blot analysis of dsRNA level after VSV infection for 12 h in HeLa cell. Methylene blue staining indicates equal RNA loading. The bar graph shows the statistics (right). $n = 3$ independent experiments. **c**, **d** Immunoblot (**c**) and dot blot (**d**) shows the immunoprecipitated (IP)-IgG or IP-dsRNA, and the m⁶A level in the IP-dsRNA in shScr and shMETTL3 HeLa cells. Methylene blue staining indicates equal RNA loading. The bar graph shows the statistics (right). **e** Anti-dsRNA-RIP-qPCR analysis of the dsRNA level in VSV RNA in shScr and shMETTL3 HeLa cells. The bar graph (Right panel) shows the statistics from the mean value of each m⁶A site (Left panel). Data are representative of 2 independent experiments. *$p < 0.05$, **$p < 0.01$, ***$p < 0.001$, ****$p < 0.0001$ as determined by two-tailed unpaired Student's $t$ test (**a**, **b**, **d**, **e**-left) or two-tailed paired Student's $t$ test (**e**-right). Error bars represent mean ± SEM.

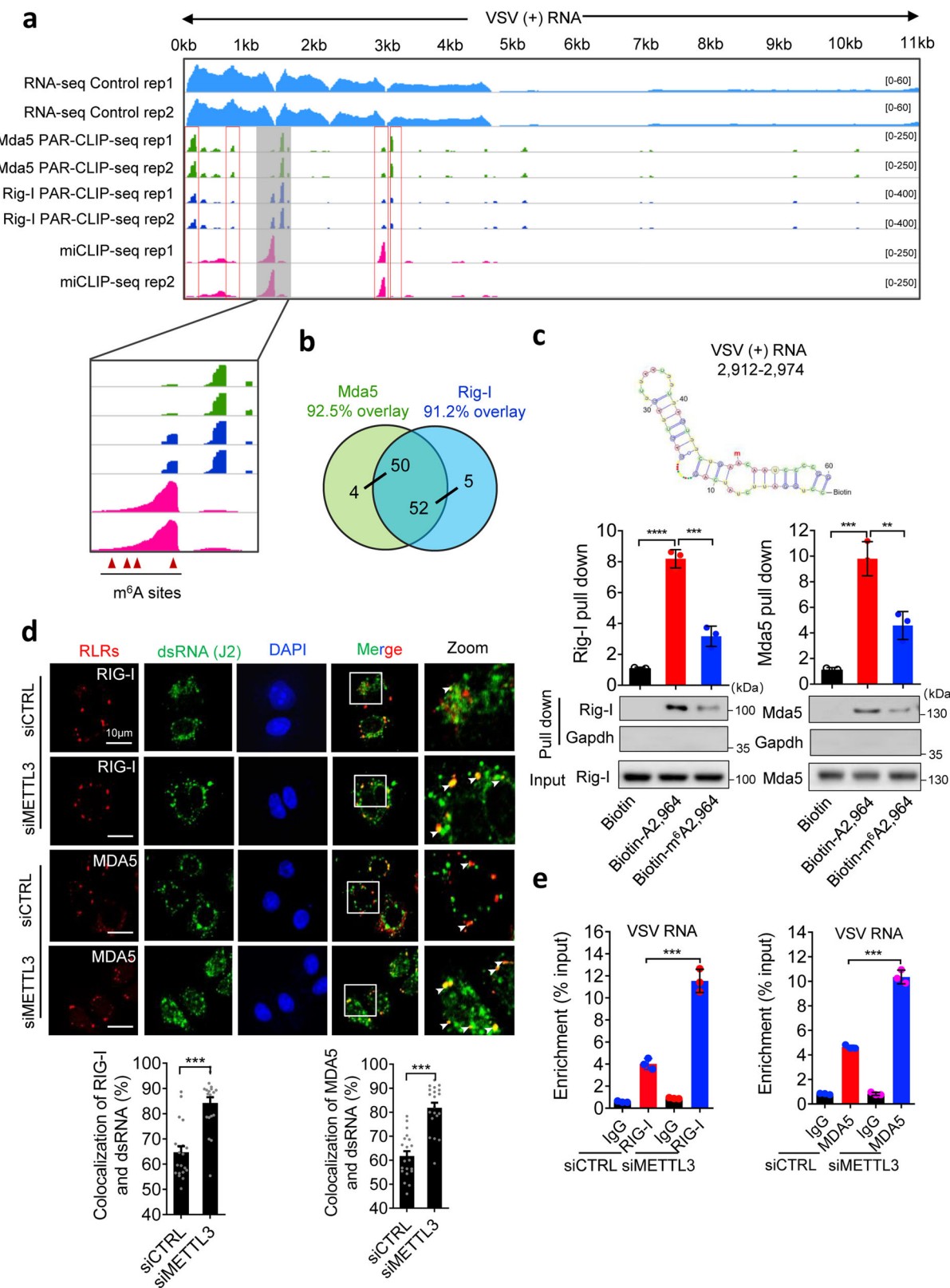

innate immune signaling through both disrupting the sensing of RLRs and impairing type I IFNs translation.

Finally, based on our investigation, targeting cytoplasmic METTL3 or METTL3 activity maybe a new strategy to ameliorate anti-viral innate immune response and cure patients who suffered from viral infections. Besides, innate immunity activation is

important for the initiation of anti-tumor immunity, therefore, it might be a potential strategy for targeting METTL3 to enhance the immunogenicity of solid tumors and promote innate immunity-induced T-cell infiltration in tumors.

In summary, our findings reveal a dsRNA structure-dependent pathway of m6A modification for controlling innate sensing

**Fig. 6 m⁶A modification impairs viral RNA sensing efficacy by RLRs. a** Integrative genomics viewer (IGV) plots the m⁶A sites and RIG-I and MDA5-binding regions on VSV (+) RNA. RNA-seq data were used as input control. **b** Venn diagram showing the overlap between high-confidence MDA5 and RIG-I binding clusters. The number of clusters in each category is shown in parenthesis. **c** Biotin-labeled RNA pull-down and Western blot analysis of Rig-I and Mda5 bindings to RNA oligo with or without single m⁶A modification (Lower graph). The upper graph indicates predicted structure of VSV (+) RNA: 2912–2974. Three times each experiment was repeated independently with similar results. Histograms show mean relative RNA content pulled down from 3 independent replicates. **d** Immunofluorescence analysis of HeLa cells increased the co-localization between RIG-I/ MDA5 and dsRNA induced by VSV infection for 8 h. $n = 20$ cells examined over 2 independent experiments with similar results. **e** RIP-qPCR analysis of increase of RIG-I and MDA5 binding to VSV (+) RNA (region: 1129–1329 nt, referred to Fig. 5a PAR-CLIP result) after deficient for METTL3 in HeLa cells. Data are representative of 2 independent experiments. *$p < 0.05$, **$p < 0.01$, ***$p < 0.001$, as determined by two-tailed unpaired Student's $t$ test (**c–e**). Error bars represent mean ± SEM.

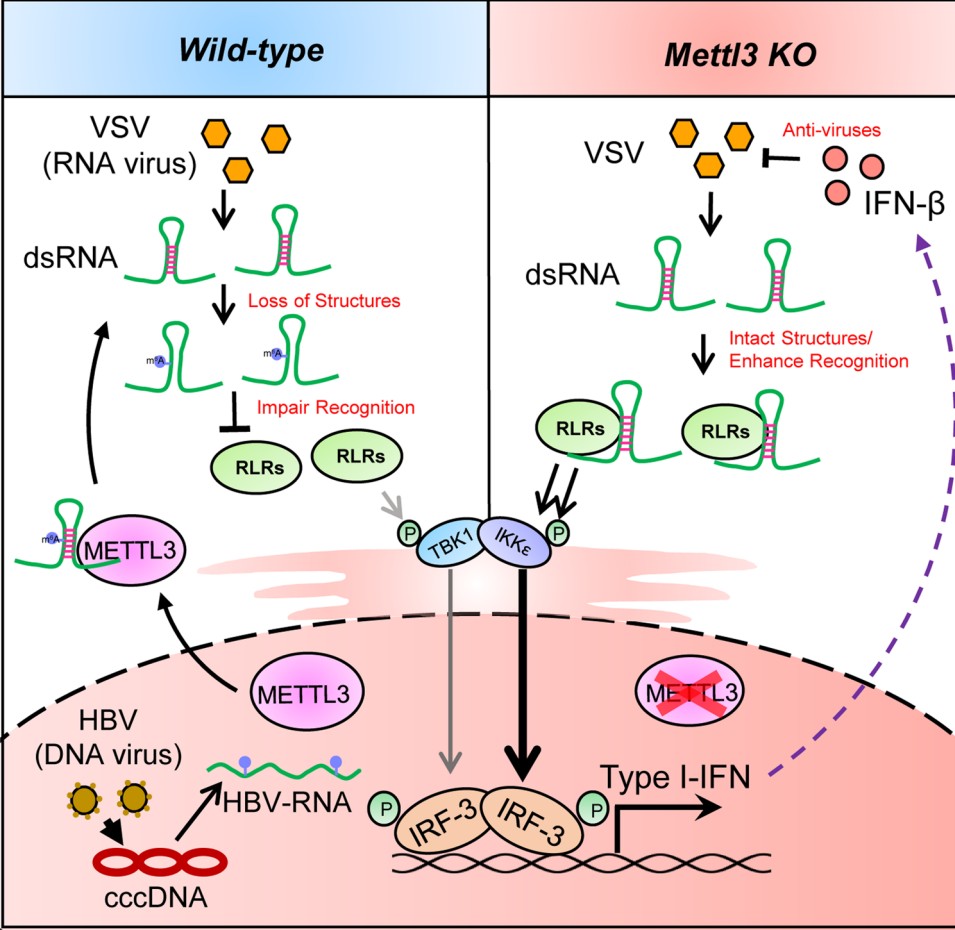

**Fig. 7 Model for METTL3-mediated m⁶A modification dampening viral RNA secondary structure to avoid innate immunity sensing.** In the proposed model, VSV RNA contains dsRNA structures to initiate innate immune sensing. During the VSV infection, METTL3 can be attracted from nucleus to cytoplasm to contact and modify VSV RNA. This m⁶A modification impairs the conformation of duplex structures in VSV RNA and interferes the sensing by dsRNA sensors involving RIG-I and MDA5, which attenuates innate immune response and helps virus invasion. When the host is deficient for METTL3, there are more dsRNA structures recognizing by RLRs to drive the expression of type I IFNs, following enhances in anti-viral function. However, the DNA virus HBV generates cccDNA in nuclear with sufficient m⁶A modification to produce less dsRNA.

system. Upon VSV infection, METTL3 is translocated to the cytoplasm to methylate viral RNAs, which decrease virus-encoded dsRNA as the mimic "self" labeling to suppress the sensing of RLRs. Therefore, we defined that m⁶A-modified viral RNA motif can act as *cis*-acting element to control innate sensing through impairing dsRNA formation and decreasing dsRNA loads.

## Methods

**Animals**. *Mettl3* floxed mice were generated by Beijing Biocytogen Co., Ltd. *Alb-Cre* mouse strain (The Jackson Lab Stock NO.: 016832) was gifted from Dr. Lijian Hui (Shanghai Institute for Biological Sciences, CAS) and *Lyz2-Cre* mouse strain (The Jackson Lab Stock NO.: 004781) was gifted from Dr. Xiyun Yan (Institute of

Biophysics, CAS). We generated *Mettl3^flox/flox^ Alb-Cre* and *Mettl3^flox/flox^ Lyz2-Cre* mice by crossing *Mettl3^flox/flox^* mice with *Alb-Cre* and *Lyz2-Cre* mice, respectively. These animals were maintained under specific pathogen-free conditions in the Animal Facilities of Institute of Biophysics, Chinese Academy of Sciences. All experimental and control mice were co-housed. And euthanasia by cervical dislocation was performed for all the animals in this study. All investigations involving mice were approved by the Animal Care and Use Committee of Institute of Biophysics, Chinese Academy of Sciences.

**Cell lines and cell culture**. RAW264.7, Vero, HEK293T, LO2, HeLa, A549, and Huh7 (Supplementary Data 2) were purchased from ATCC. AC12 cell line is gifted from Dr. Wenhui Li (National Institute of Biological Sciences, Beijing). All these cell lines were maintained in high-glucose DMEM (Gibco) supplemented with 10% FBS (Gibco) and 100 U/mL penicillin, 100 μg/mL streptomycin. The cells were incubated at 37 °C in a humidified chamber containing 5% $CO_2$.

**Transfection**. 2 µg DNA of each plasmid (Supplementary Data 4) or poly (I:C) (InvivoGen, CAT# tlrl-pic) or poly (dA: dT) (InvivoGen, CAT# tlrl-patn) was used to transfect cells with Lipofectamine 2000 (Life Technologies) or GenStar (C101-01) transfection reagents, according to the manufacturers' instruction. The gene expression level was analyzed by qRT-PCR or Western blot.

**RNA interference**. Cells were transfected with 100 nM siMETTL3 or siScramble, respectively, using RNAi Max (Invitrogen) for 48 h according to the manufacturer's protocols. The genes expression level was analyzed by qRT-PCR or Western blot. The sequence information of siRNAs was provided in the Supplementary Data 3.

**CRISPR/Cas9**. The guide-RNA oligo, with sequence for its target gene, was designed using crispr.mit.edu website. The sequence was cloned into px458M-Cas9-2A-EGFP-MCS vector. RAW264.7 cells were transiently transfected with px458M-Cas9-2A-EGFP-MCS plasmid with sgRNA. Cultured about 2 days, the EGFP-positive single cells were sorted into the 96-well plate (1 cell/well) by Aria III cell sorter. The knock out efficiency of colonies were detected by Western blot. And the sequence information of sg*Mettl3* was provided in the Supplementary Data 3.

**Western blot**. Total proteins were lysed from cells by RIPA buffer (150 mM NaCl, 1%NP40, 0.5% sodium deoxycholate, 0.1% SDS, 50 mM Tris pH 8.0 and 1 mM phenylmethylsulfonyl fluoride) containing a protease inhibitor cocktail and a phosphatase inhibitor on ice for 30 min. The concentration of proteins was detected with BCA Protein Assay Kit (Pierce, Cat#23225) and samples with loading buffer were boiled at 100 °C for 3 min. Each sample was separated into SDS-PAGE and transferred to PVDF membranes. The PVDF membrane was blocked with 5% non-fat milk and incubated with specific primary antibodies (working concentration of each antibody refers to product specification) overnight. Then the membrane was probed with appropriate secondary antibody, and detected the protein bands with Immobilon™ Western HPR Substrate Luminol Regeant (Merck Millipore, Cat#WBKLSO500). Antibody information (Supplementary Data 1).

**RNA extraction and real-time RT-PCR**. Total RNAs extracted from cultured cells were used for qRT-PCR analysis. Briefly, cells in culture were lysed in TRIzol (Ambion, Cat#15596018) and RNA extraction was performed according to the manufacturer's instruction. Total RNA was reversely transcribed into cDNA with PrimeScript TM RT Master Mix (TaKaRa, Cat#RR0364). Reverse transcription was performed at 37 °C for 45 min, followed at 80 °C for 5 s. The cDNA samples were diluted and stored at 4 °C. Quantitative real-time PCR (qRT-PCR) analysis was performed with TB Green™ PremixEx Taq™ (TaKaRa, Cat#RR420A) on the ViiA 7 Real-Time PCR system. All runs were accompanied by the internal control genes *Actin* for murine and *ACTIN* for human. And all the primer sequences were provided in the Supplementary Data 3.

**ELISA**. The ELISA assay was performed with a Mouse IFN-β ELISA Kit (BioLegend, Cat#439407) (Supplementary Data 5) according to the manufacturer's instructions.

**Immunofluorescence staining**. Cells were fixed with 4% paraformaldehyde (PFA) for 30 min at room temperature (RT) and then permeated with 0.1% Triton X-100 for 20 min on ice. Next, cells were blocked with 1% BSA for 1 h at RT, following incubation with specific primary antibodies overnight. Next day, the samples were incubated with appropriate secondary antibodies at RT for 2 h, and DAPI was performed for nuclear staining. Cells were visualized with the Zeiss LSM700 laser-scanning confocal microscope. And all the data were analyzed by ZEN2012 software.

**Luciferase assay**. The promoter region of the IFN-β gene was cloned into the pGL3-basic plasmid. The recombinant plasmid with or without pcDNA3.1-Flag-METTL3 co-transfected into HEK293T cells. Then treated the cells with or without VSV or other triggers for 6 h post transfection of 24 h. Firefly and Renilla luciferase activity were measured with a Dual-Luciferase reporter system (Promega) according to the manufacturer's protocol. Data were normalized by the activity of Renilla luciferase.

**H&E staining**. 4% paraformaldehyde fixed the tissues, and then paraffin embedded tissue sections and mounted on glass slides. These slides were used for H&E staining, deparaffinized and then stained with Hematoxylin and Eosin (YEASEN), respectively. Finally, mounted the tissues with resins and observed using microscope.

**Dot blot**. mRNA was purified from total RNA through selectively depleting ribosomal RNA (rRNA) using RiboMinus™ kit (Thermo Fisher). Equal amounts of mRNA were spotted to a nylon membrane (Fisher) by Bio-Dot™ apparatus (BIO-RAD), and then irradiated by UV crosslinking at UV 254 nm with 0.12 J/cm². After blocking in TBST buffer containing 5% non-fat milk for 1 h, the membrane was incubated with 1:2000 diluted anti-m⁶A or 1:500 diluted J2 antibody overnight

at 4 °C. The membrane was washed with TBST and then incubated with HRP-conjugated secondary antibody (1:5000) for 2 h and visualized by using Immobilon™ Western HPR Substrate Luminol Regeant (Merck Millipore, Cat#WBKLSO500).

**UV-crosslinking RNA IP**. Two 15-cm dishes HeLa cells infected with VSV for 12 h were washed twice with 6 ml cold DEPC-PBS and irradiated at 150 mJ/cm² at 254 nm in a Stratalinker. Cells were collected into a 15 ml tube and resuspended with 1 ml UV crosslinking RIP buffer (50 mM Tris-HCl pH7.5, 150 mM NaCl, 1% NP40, 0.5% sodium deoxycholate, 1 mM PMSF, 2 mM VRC, protease inhibitor cocktail). After sonication on ice with 10 s ON, 10 s OFF for 3 min, cells were collected by centrifugation at $12,000 \times g$ for 10 min at 4 °C. The supernatants were collected into a 15 ml protein A/G beads and 20 µg/ml yeast tRNA at 4 °C for 30 min. Afterwards, the supernatants were collected to incubate with the primary antibody and pre-coated protein A/G beads overnight at 4 °C. After incubation, the beads were first washed with washing buffer I (50 mM Tris-HCl pH7.5, 1 M NaCl; 1% NP40, 1% sodium deoxycholate, 2 mM VRC) 5 min for 3 times, then with washing buffer II (50 mM Tris-HCl pH7.5, 1 M NaCl, 1% NP40, 1% Sodium Deoxycholate, 2 mM VRC, 1 M urea) 5 min for 3 times, and resuspended with 140 µl elution buffer (100 mM Tris-HCl pH7.0, 5 mM EDTA, 10 mM DTT, 1% SDS). 40 µl was saved for protein analyses and 100 µl for RNA extraction. 5 µl of 10 mg/ml proteinase K was added into the RNA sample and incubated at 55 °C for 30 min. Finally, 1 ml TRIzol (Ambion) was added for total RNA isolation and detected by qRT-PCR.

**Biotin-labeled RNAs to pulldown proteins**. Biotin-labeled RNA pulldown assay was performed as described[38]. Biotin-poly(A:U), biotin-poly(m⁶A:U), biotin-A1272, and biotin-m⁶A1272 were artificially synthesized (Sangon Biotech). The sequences information was provided in the Supplementary Data 3. $2 \times 10^7$ RAW264.7 cells were treated with 200 µl cell lysis (150 mM NaCl, 1 mM EDTA, 1% Triton X-100, 0.5% DTT, 50 mM Tris-HCl, 50 mM Tris-HCl, pH 7.5, 0.5% sodium deoxycholate, 1 mM PMSF, and 2 mM VRC, with protease inhibitor cocktail, Roche and 2.5 µl RNasin ribonuclease inhibitor, Promega) on ice for 30 min. The supernatant was collected to a new tube, pre-cleared 40 µl Streptavidin Dynabeads (Invitrogen) was added for 30 min at 4 °C, then 20 µg/ml yeast tRNA was added to block unspecific binding and incubated for 20 min at 4 °C. 4 µg biotinylated RNAs were added for incubation 1.5 h at RT and then by addition of 40 µl Streptavidin Dynabeads to incubate for 2 h at RT. Beads were washed 5 min for 4 times with RIP buffer containing 0.5% sodium deoxycholate and then saved 40% for running agarose gel and boiled 60% sample in 2× SDS loading buffer at 100 °C for 5 min. The RNA–protein complex was detected by Western blot with primary antibodies of anti-MDA5 and anti-RIG-I.

**Viral infection and plaque assay**. Mouse peritoneal macrophages or other cells were seeded for 24 h before virus infection. Cells were infected with VSV, HSV-1, HCMV, HCV, or SeV for various times, as indicated in the figures. VSV plaque assay and VSV replication were determined by a standard TCID₅₀ assay on permissive Vero cell monolayers in 96-well plates with a series of tenfold-diluted samples. After 1 h of infection, the plates were incubated for 48 h. The medium was removed and the cells were fixed with 4% paraformaldehyde for 30 min and stained with 1% crystal violet for 30 min before plaque counting.

**Viral infection in vivo**. For in vivo VSV infection studies, 6-week-old control and monocyte or hepatocyte-specific METTL3-depleted mice were infected with high dose of VSV ($1 \times 10^9$ PFU/mouse) or moderate dose of VSV ($2.5 \times 10^8$) by tail vein injection. High dose of VSV induced mice death in a short time to detect the resistance of mice for VSV infection. For moderate dose of VSV infected mice, 24 h after infection, we collected the blood from the orbital sinus for ELISA and obtained the lungs, spleen, and liver from each mouse for analysis of RNA and protein. The liver and lung were fixed by 4% paraformaldehyde for H&E staining.

**RNA-seq and miCLIP-seq**. mRNAs were purified from total RNAs using Dynabeads mRNA purification kit (Life Technologies, 61006), and subjected to generate the cDNA libraries according to TruSeq RNA Sample Prep Kit protocol. All samples were sequenced by Illumina HiSeq X-ten with paired-end 150 bp read length.

Single-base resolution high-throughput sequencing was carried out according to previously reported methods with some modifications[39,40]. Briefly, mRNAs were purified using Dynabeads mRNA Purification Kit (Life Technologies, 61006) and fragmented to a size around of 100 nt using the fragmentation reagent (Life Technologies, AM8740). 10 µg of purified mRNAs were mixed with 25 µg of anti-m⁶A antibody (Abcam, ab151230) in 450 µl immunoprecipitation buffer (50 mM Tris, pH 7.4, 100 mM NaCl, 0.05% NP-40) and incubated by rotating at 4 °C for 2 h. The solution was then transferred to a clear flat-bottom 96-well plate (Corning) on ice and irradiated three times with 0.15 J/cm² at 254 nm in a CL-1000 Ultraviolet Crosslinker (UVP). The mixture was then immunoprecipitated by incubation with Dynabeads Protein A (Life Technologies, 1001D) at 4 °C for 2 h. After extensive washing and on-bead end-repair and linker ligation, the bound RNA fragments were eluted from the beads by proteinase K digestion at 55 °C

for 1 h. RNAs were isolated from the eluate by phenol-chloroform extraction and ethanol precipitation. Purified RNAs were reversely transcribed with Superscript III reverse transcriptase (Life Technologies, 18080093) according to the manufacturer's protocol. First-strand cDNA was size-selected on a 6% TBE-Urea gel (Life Technologies, EC6865BOX), and circularization and re-linearization of cDNA were performed with CircLigase II (Epicentre, CL9021K) and BamHI (NEB, R0136), respectively. Libraries were PCR amplified with Accuprime Supermix 1 enzyme (Life Technologies, 12342010) for 20 cycles and size-selected on an 8% TBE gel (Life Technologies, EC6215BOX). Sequencing was carried out on Illumina HiSeq X-ten platform according to the manufacturer's instructions.

**PAR-CLIP.** RAW264.7 cells treated with VSV for 12 h were cultured in medium supplemented with 200 μM 4-thiouridine (4-SU) (Sigma) for 14 h, and then irradiated once with 400 mJ/cm² at 365 nm using the CL-1000 Ultraviolet Crosslinker (UVP) for crosslinking. Cells were harvested in lysis buffer (50 mM Tris-HCl pH 7.5, 100 mM NaCl, 2 mM EDTA, 0.5% (v/v) NP-40, 1 mM NaF, 1× protease inhibitor cocktail (Bimake), 0.04 U/ml RNasin (Beyotime)) and rotated for 30 min at 4 °C. Cell debris was removed by centrifugation at 12,000 × g for 30 min at 4 °C and the supernatant (3–4 mg/ml) was digested by 1 U/μl RNase T1 at 22 °C in a water bath for 8 min and cooled on ice for 5 min. Then the lysates were incubated with indicated antibody overnight at 4 °C. Washed magnetic Protein A beads were added to the mixture and incubated for another 2 h, and the beads were then washed three times with IP wash buffer (50 mM Tris-HCl pH 7.5, 300 mM NaCl, 0.05% (v/v) NP-40, 1× protease inhibitor cocktail (Bimake), 0.04 U/ml RNasin (Beyotime)). Beads were digested with 10 U/μl RNase T1 again at 22 °C in a water bath for 8 min, cooled on ice for 5 min, then washed three times in high salt wash buffer (50 mM Tris-HCl pH 7.5, 500 mM NaCl, 0.05% (v/v) NP-40, 1× protease inhibitor cocktail (Bimake), 0.04 U/ml RNasin (Beyotime)), resuspended in 100 μl dephosphorylation buffer (50 mM Tris-HCl pH 7.9, 100 mM NaCl, 10 mM MgCl₂), and incubated with 0.5 U/μl calf intestinal alkaline phosphatase (CIP, NEB) for 10 min at 37 °C with gentle rotation. Beads were then washed twice with phosphatase wash buffer (50 mM Tris-HCl pH 7.5, 20 mM EGTA, 0.5% (v/v) Triton X-100) with 3 min rotation. Beads were resuspended with 200 μl proteinase K buffer (100 mM Tris-HCl pH 7.5, 50 mM NaCl, 10 mM EDTA, 4 μg/μl proteinase K (Roche)), and incubated at 55 °C for 2 h. RNA was extracted with Phenol: Chloroform. Sequencing libraries were constructed using the SMARTer smRNA-seq kit (Clontech) according to the manufacturer's instructions. Sequencing was performed on an Illumina HiSeq X-Ten instrument with paired end 150-bp read length.

**RNA-seq analysis.** The quality of raw sequencing data was checked by FastQC (v0.11.5) first. Genomic alignment was performed on paired-end reads using hisat2 (v2.0.5)[41] aligner to mice reference genome (GRCm38/mm10; Ensembl version 68) and VSV positive-sense (+) RNA (RefSeq, NC_001560), respectively, with the default settings. Only reads with mapping quality score (MAPQ) ≥ 20 were kept for the downstream analysis. Then, FeatureCounts (v1.6.0)[42] was employed to calculate the read counts of per gene. We also obtained the Reads Per Kilobase per Million mapped reads (RPKM) of genes in each sample based on FeatureCounts (v1.6.0)[42] results with custom R scripts. Differentially expressed genes between samples were identified by R-package DESeq2 (v3.18.1)[43] with |fold change| >1.5 and p-value < 0.05 as thresholds. Meanwhile, we kept genes present (RPKM > 1) in all samples for further analysis. Gene ontology (biological process) and Reactome pathway enrichment analysis was performed using web-based tool ToppFun from ToppGene Suite (https://toppgene.cchmc.org/)[44]. The terms or pathways with p-value < 0.05 were considered significant. Gene set enrichment analysis was carried out by GSEA (http://www.broad.mit.edu/GSEA) software with immunologic signatures gene sets (c7.all.v6.2.symbols.gmt).

**miCLIP-seq analysis**

*Read processing.* miCLIP-seq data analysis pipeline was similar to previously described[39]. Adaptor sequence were removed by fastx_clipper tool from FASTX-Toolkit (http://hannonlab.cshl.edu/fastx_toolkit). The forward reads were demultiplexed based on barcode by fastq2collapse.pl from CTK Tool Kit (v1.0.3)[45] to remove PCR-amplified reads, then, Cutadapt (v1.16) was employed to trim the polyA-tail on each sample. Reverse reads were reversely complemented by using fastx_reverse_complement tool from fastx_toolkit and processed in the same way. Finally, reads were subjected to stripBarcode.pl from CTK Tool Kit (v1.0.3)[45] to remove the random barcode, followed by the short reads (<18nt) removal by Trimmomatic (v0.33)[46].

*Mapping and mutation calling.* Since miCLIP-seq technique is sensitive to sequencing depth, we pooled the remained reads together from the replicate samples and aligned to mice reference genome (GRCm38/mm10; Ensembl version 68) and VSV positive-sense (+) RNA (RefSeq, NC_001560), respectively, by BWA (v0.7.17-r1188)[47] with the recommended parameter: -n 0.06 -q 20. To determine the m⁶A sites, CTK Tool Kit (v1.0.3)[45] was performed to detect cross-linking-induced mutation sites (CIMS) in miCLIP-seq data as reported[48]. For each mutation position, the CIMS software identifies the coverages of unique tags ($k$) and mutation position ($m$). To filter background mutations, we only kept the sites

with an $m/k$ ratio 1–50% and mutation positions within the RRACH motif as reliable m⁶A sites for subsequent analysis[49,50]. The m⁶A motif was generated by WebLogo3[51]. For the IGV plots, genomeCoverageBed tool from BedTools (v2.26.0)[52] was used to transform alignments into bedGraph file with the scaled RPM (1,000,000/mapped reads) for mouse genome, and (1000/mapped reads) for VSV, UCSC bedGraphToBigWig tool (http://hgdownload.cse.ucsc.edu, v4) was employed to convert into bigwig format file, in order to uncover the m⁶A distribution along different VSV RNAs, the reads that mapping to VSV genomic (−) RNA or VSV positive-sense (+) were distinguished, respectively. Finally, Integrative Genomics Viewer (IGV)[53] were used to visualize the distributions of the m⁶A modification.

**PAR-CLIP-seq analysis.** Forward sequencing reads were trimmed by Cutadapt (v1.16) to remove low quality bases and adapters, and then aligned to the mice reference genome (GRCm38/mm10; Ensemble version 68) and VSV positive-sense (+) RNA (RefSeq, NC_001560), respectively, using bowtie(v1.0.1)[54] with following parameter: -v 2 -m 10 --best -strata[55]. Only the reads short than 100nt were kept for peak calling. Binding regions of RBPs were obtained by PARalyzer (v1.5)[56] with default settings, a software defined the binding cluster based on T-to-C conversions. We filtered the binding region with reads count ≥50 for more analysis.

**Statistics.** Unless otherwise indicated, data are presented as mean ± SEM of 3 independent experiments. All statistical analyses were performed with Graph Prism 6.0 software, and the statistics were analyzed by unpaired Student's t test. The correlation between genes expression was analyzed using Pearson's test. Survival rate and overall patient survival were analyzed by Kaplan–Meier survival curve. p values were provided as *$p < 0.05$, **$p < 0.01$, ***$p < 0.001$, and ****$p < 0.0001$.

**Reporting summary.** Further information on research design is available in the Nature Research Reporting Summary linked to this article.

## Data availability

The raw data can be accessed from Genome Sequence Archive at the National Genomics Data Center, Beijing Institute of Genomics, CAS/China National Center for Bioinformation as accession CRA002259 [https://bigd.big.ac.cn/search/?dbId=gsa&q=CRA002259]. Besides, the original data is also available at Sequence Read Archive with number PRJNA634708 [https://www.ncbi.nlm.nih.gov/bioproject/PRJNA634708/]. The raw numbers for charts and graphs are available in the Source Data file whenever possible. All other data supporting the findings of this study are available from the corresponding author on reasonable request. Source data are provided with this paper.

## Code availability

The custom R scripts for RPKM calculation is available on GitHub [https://github.com/zqyhyunbin/VSV-m6A]. Other custom scripts for analyzing data are available upon request.

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

## Acknowledgements

We thank Prof. Wenhui Li and Prof. Ailong Huang for HepG2-NTCP (AC12 clone) and HepAD38 cell lines, Prof. Lijian Hui and Prof. Xiyun Yan for Alb-Cre and Lyz2-Cre mice, and all members in Yang lab and the Key Laboratory of Infection and Immunity of CAS at the Institute of Biophysics for their help and suggestions. This work was supported by grants: Chongqing International Institute for Immunology (No. 2020YJC02), The Strategic Priority Research Program of the CAS (No. XDB29040102, XDB37030206, XDB37030205), National Science and Technology Major Project of China (ZX10301404-002-001), National Natural Science Foundation of China (No. 81630069, 81672464, 81522030, 81773262, 81902509), the National Natural Science Fund for Distinguished Young Scholars (31625016); The Open Project Program of the State Key Laboratory of Cancer Biology (CBSKLZDKF01), CAS Key Research Projects of the Frontier Science (QYZDY-SSW-SMC027, KJZD-SW-L05); the Youth Innovation Promotion Association, CAS (2018133); the K. C. Wong Education Foundation, and the NSFC consulting grant (91953000).

## Author contributions

W.Q. designed and performed most of the experiments, data analysis and wrote the manuscript. Q.Z. did most of the bioinformatics analysis. Y.L., Z. G., X. Y., G. C., and X. W. did the experiments. R.Z., H.T., Y.P., H.D., H. P., and A.Y. developed the methodology. W.Q., Y. Y., B. S., Y. G., Z. Y., and X. W. revised the manuscript. P.Y. and Y. G. Y. initiated, designed the project, wrote, and revised the manuscript.

## Competing interests

The authors declare no competing interests.
