## [Peer Review File · Nature Communications]

REVIEWER COMMENTS

Reviewer #1 (m6A modification, immune signaling.) (Remarks to the Author):

This is a very extensive and well done study studying the role of Mettl3 and m6A on in regulating response to VSV RNA virus infection in mice. The authors convincingly show that VSV infection is better handled when Mettl3 is removed from host macrophages which then can better mount anti viral response early on and better RNA virus sensing. The authors show that upon infection, Mettl3 is partially translocated to cytoplasm which then can modify viral transcripts and alter their secondary structure and then leading to reduced sensing by RNA virus cytoplasmic sensors.

The study is massive and novel. cutting edge conditional knockout mice, sequencing techniques etc are used throughout. The conclusions are solid, robust and well supported by the high quality data in the paper. Discussion is informative and references are adequate throughout the manuscript.

Minor comments:

- the authors indicate that they do not see Mettl14 translocation to cytoplasm upon VSV infection but only for Mettl3. Given that it is well established that both Mettl3 and Mettl14 are needed in the writing complex, how can the authors then explain that viral RNA are being methylated by Mettl3 without Mettl14 around nearby ?

- although not necessary, can the authors show some minimal data (in vitro can be sufficient) on another RNA virus beside VSV ?

Reviewer #2 (RNA virus, innate immunity.) (Remarks to the Author):

Review: N6-methyladenosine RNA modification suppresses antiviral innate sensing pathways via reshaping double-stranded RNA modules (Qiu et al., 2020)

In this study, Qiu et al. propose that N6-methyladenosine RNA modification of viral dsRNA serves as a major suppressor of the innate immune sensing signalling cascade. They use a combination of transient overexpression and knock down of the m6A methylase METTL3 in a range of cell lines and find that not only is the type I IFN response suppressed by METTL3 but also innate immune sensing upstream of IFNB transcription in response to VSV infection. Using knockout of METTL3 in monocytes and hepatocytes in a mouse model, they were able to confirm these observations in vivo, demonstrating reduced viral titres and increased type I IFN response after viral challenge. Furthermore, they show that METTL3, usually found in the nucleus of cells, translocates to the cytoplasm and co-localises and interacts with viral RNA as well as m6A RNA. Concurringly, they find that knock down of METTL3 corresponds with higher levels of dsRNA and lower levels of m6A RNA in infected cells. Finally, using PAR-CLIP and miCLIP sequencing techniques, they find that although both RIG-I and MDA5 sense and bind to VSV RNA, regions of high binding corresponded to regions with low levels of m6A modification, and regions with high levels of m6A modification corresponded to regions with poor interaction to RIG-I and MDA5. They conclude that METTL3-mediated m6A modification of viral RNA remodels viral dsRNA structures, preventing effective recognition by RLRs and suppressing the type I IFN response to viral infections. Overall, this study provides a novel insight into how detection of viral RNA structures is regulated by PRRs like RIG-I and MDA5 and how there are intrinsic negative regulators of viral sensing. However, despite the overall value of this study to the field, there are some major and minor issues with the manuscript that need to be addressed.

Major issues:

1. The manuscript fails to meet acceptable standard of English language and is in need of professional proof reading and editing prior to resubmission. At times, unintentional wrong choice of words leads to sentences actually stating the opposite of what the authors presumably intend to say (e.g. lines 41-44: "The innate immune system is highly conserved among plants and animals, which recognizes the invading pathogens through pattern recognition receptors (PRRs) to trigger an effective immune response for defending the pathogens.").
2. For IP experiments (RIG-I / MDA5) no control protein was pulled down. It is commonly known that many proteins have random RNA-binding ability. To exclude that these specific RNA-binding profiles of viral RNA are not simply random background binding, these experiments would require a negative control protein for the Ips (Figure 4f, Figure 6a).
3. The authors make claims that the dsRNA 'structure' is significantly altered by m6A modification. While this is a reasonable hypothesis, they do not offer any actual structural or biochemical data supporting this hypothesis.

Minor issues:

1. Why did this study focus on using murine macrophages for many of the studies? In general, experiments were spread across murine macrophages, HEK-293T cells (whose suitability to study the innate immune response is dubious at best) and HeLa cells.
2. Figure 1d/e/f/g: p-IRF3 Western blots are mostly not very convincing in supporting the claims made in the text.
3. Figure 1i: there is no control where METTL2 isn't overexpressed and where IRF3 translocation isn't blocked.
4. Figure 1i (graph): colours aren't defined (red / black) and it is not clear where the images of those control cells are (not shown).
5. Figure 1l: the text (lines 124-126) mentioned up- and downregulated genes in infected cells METTL3-depleted cells compared to mock infected cells. However the figure panel shows RPKM values for infected KO vs WT cells. An alternative way of visualizing the data might be more accurate or acceptable, for example volcano plots taking p values into account and depicting fold change over mock infected cells.
6. Figure 2b and 2c: It's not immediately apparent what we are looking at and whether this image is in any way representative of the phenotype. Is there any way to quantify these images to get a sense of how valid the authors' claims are?
7. Figure 2d: The statistics on the error bars especially for Liver and Lung Ifnb1 mRNA levels looks dubious considering the variance of datapoints.
8. Line 154: the authors claim "resistance" to VSV-induced lethality although Figure 2i shows that there was some mortality.
9. Lines 169-174: 'other' treatments are mentioned, however only a DNA virus (HSV) and poly(I:C) and poly(dA:dT) treatment is shown at a single timepoint. I would hesitate to make the authors' statement that this is specific to VSV infection without testing this in other cytoplasmic and nuclear RNA viruses as well as poly(I:C)/poly(dA:dT) treatment at various timepoints. It is likely that, since different viruses and treatments have different kinetics, that by simply looking at a single time point different phenotypes are missed.
10. Lines 212-216: It is not clear to me how the finding that specifically only positive-sense viral RNA was associated with METTL3 supports their hypothesis that VSV infection triggers cytosolic translocation of METTL3 to interact with VSV RNA.
11. Figure 6b is not mentioned in the main text.

Reviewer #3 (Innate signaling, IFN.) (Remarks to the Author):

The manuscript entitled "N6-methyladenosine RNA modification suppresses antiviral innate sensing pathways via 1 reshaping double-stranded RNA modules" by Qiu et al. described a potential role of the m6A methyltransferase METTL3 in suppressing Type I interferon response by reducing the sensing efficacy of RIG-I and MDA5 to double strand RNA.

Overall, the work presented is high-quality, the findings are novel and most conclusions are well-supported by experimental evidence. However, the mechanisms for regulating the METTL3 activity and the specificity of METTL3 involving type I interferon responses to VSV versus other viruses or RNA versus DNA are not clear. The manuscript also needs to be edited and revised, because there are significant issues with usage and grammar that interfere with meaning and readability.

In figure 1a, METTL3 overexpression blunts IFN β 1 response to several stimuli/infection conditions. However, the mechanisms proposed in the rest of the manuscript such as the effects of METTL3 on dsRNA formation could not explain the observation that METTL3 overexpression inhibits IFN β 1 response to poly dA:dT, LPS and DNA viruses. What effect does METTL3 overexpression have on IRF3 phosphorylation upon infection with another RNA virus? DNA virus? Stimulation with RNA or DNA analog?

It would be good to see the equivalent of figures 1e-h some of these conditions, (and/or a modified supplemental figure 4b) because the rationale for focusing solely on VSV after this is unclear. Figure 3b and 3c present a clear result: endogenous METTL3 moves from the nucleus to the cytoplasm in VSV infection. These and supplemental figure 4b are referred to in the text (line 169-170) as evidence that METTL3 doesn't move to the cytoplasm in conditions other than VSV infection, but S4b uses overexpressed METTL3 rather than endogenous METTL3. That's fine for supplemental figure 1j, which shows that METTL3 overexpression stops IRF-3 nuclear translocation in VSV infection, but it's not valid evidence for comparison to 3b/c to make the VSV-specific claim. Western blots as in 3b would be best. On the other hand, the addition of an IRF3 staining (as in supplemental figure 1j) would make supplemental figure 4b much more useful and could effectively concerns with these figures.

The model the authors present is not solid without data to show the effect of NLS mutant overexpression on IRF3 translocation. Is cytoplasmic METTL3 alone not sufficient for the loss of nuclear IRF3 in non-VSV infection conditions (or stimulation conditions) or is the presence of VSV also required?

As the authors acknowledge, there isn't any direct evidence presented of a virus-originated mechanism for METTL3 translocation or dsRNA methylation in VSV infection. However, catalytic METTL3 is known to complex with METTL14, which recognizes methylation targets. Is there a viral protein acting as a cytoplasmic partner for METTL3? The manuscript would be better if the authors checked for physical interaction between METTL3 and VSV proteins, and VSV-L is particularly interesting as a possible partner given its role in VSV's atypical mRNA capping and SAM-dependent methylation. This might be discussed as a future direction.

A few minor points:

- On lines 102-108, the explanation of how these results differ from the previous report of METTL3 impact on IFN β 1 could be more clear.
- Given the adverse effect of m6A modification on translation rate (Slobodin et al. in Cell, 2017), is there a tradeoff involved in this apparent viral strategy?

- As another point of discussion, do the author's findings regarding the m6A modification-induced blunting of antiviral response indicate any application potential (ie mRNA delivery into cells?)
- The term "respectively" is used unnecessarily several times. It's used correctly on lines 49 and 559.
- In Figure 7, it looks like "less structures" should be "loss of structure." "Keep structures" might be better as "intact structure."

Revision Summary

1. Summary of major comments from Editor and Reviewers.

Based on the requests and comments from the reviewers, we have performed further analyses and designed additional experiments. The details of our revision results are summarized and listed in the following **Table 1**.

Table 1. Revision results for major comments from the reviewers.

Key Questions	Reviewers	Clarification on the original submission data	Performed new experiments	New supporting data
1 The manuscript fails to meet acceptable standard of English language and is in a need of professional proof reading and editing prior to resubmission.	2#: Q1 3#: Minor issues	N/A	Careful proofreading and checked by an English native speaker.	N/A
2 Negative control (IgG) for the (RIG-I / MDA5) IP.	2#: Q2	In the original Figure 6e , we introduced IgG as the negative control when performing the IP experiments (RIG-I/MDA5) and found that IgG did not effectively pull down VSV RNAs.	We have performed western blot using the indicated antibodies for the IP experiment and showed the specificities.	Our new data indicated the great binding specificities of these antibodies. (Rebuttal Figure 2, Revised Figure S5d and S7b)
3 Any actual structural or biochemical data could support the hypothesis that m ⁶ A modification alters dsRNA 'structure'?	2#: Q3	The original Figure 4e and Figure S7e results suggest that m ⁶ A modification impairs dsRNA structures on VSV RNA.	We have conducted IF for dsRNA in WT and METTL3 knockdown HeLa cells.	Our new data showed that METTL3 knockdown increased dsRNA formation upon VSV infection (Rebuttal Figure 3, Revised Figure 5a). We also changed the description and included citation in the discussion part.
4 Using another way to visualize the DEGs in infected cells METTL3-depleted cells compared to mock infected cells	2#: Minor issues (5)	N/A	We have added the volcano plot, which could make the results much more accurate.	Our new data indicated that only 498 genes changed and nearly no ISGs changed by comparing WT and KO in mock infection (Rebuttal Figure 6, Revised Figure S2a).
5 Different viruses and treatments have different kinetics, that by simply looking at a single time point different phenotypes are missed.	2#: Minor issues (9)	N/A	We have carried out IF for METTL3 upon different treatments in various timepoints.	Our new data indicated that only RNA viruses SeV and VSV induced METTL3 cytosolic translocation (Rebuttal Figure 8, Revised Figure 3d and Figure S4b).
6 What effect does METTL3 overexpression have on IRF3 phosphorylation upon infection with another RNA virus? DNA virus? Stimulation with RNA or DNA analog?	3#: Q1	N/A	We have performed dual-luciferase reporter assay to detect the activation of IRF3 upon infection with another RNA virus, DNA virus or stimulation with RNA or DNA analog.	Our new data revealed that SeV and VSV-induced IRF3 activation were suppressed by METTL3 overexpression (Rebuttal Figure 9, Revised Figure 1b).
7 Is cytoplasmic METTL3 alone not sufficient for the loss of nuclear IRF3 in non-VSV infection conditions (or stimulation conditions) or is the presence of VSV also required?	3#: Q3	N/A	We have performed dual-luciferase reporter assay to detect the activation of IRF3 upon non-VSV infection conditions or stimulation conditions.	SeV or VSV-induced IRF3 activation was also suppressed by METTL3-NLS-mutant overexpression (Rebuttal Figure 10, Revised Figure 3i).

2. Point-by-point responses to reviewers

Reviewer #1

This is a very extensive and well done study studying the role of Mettl3 and m6A on in regulating response to VSV RNA virus infection in mice. The authors convincingly show that VSV infection is better handled when Mettl3 is removed from host macrophages which then can better mount anti viral response early on and better RNA virus sensing. The authors show that upon infection, Mettl3 is partially translocated to cytoplasm which then can modify viral transcripts and alter their secondary structure and then leading to reduced sensing by RNA virus cytoplasmic sensors.

The study is massive and novel. cutting edge conditional knockout mice, sequencing techniques etc are used throughout. The conclusions are solid, robust and well supported by the high quality data in the paper. Discussion is informative and references are adequate throughout the manuscript.

Response: We thank the reviewer's approval of our work. We have performed a series of additional experiments to show METTL3 translocation on another RNA virus infection. The detailed evidence has been included in the following point-to-point responses.

Minor comments:

1). *The authors indicate that they do not see Mettl14 translocation to cytoplasm upon VSV infection but only for Mettl3. Given that it is well established that both Mettl3 and Mettl14 are needed in the writing complex, how can the authors then explain that viral RNA are being methylated by Mettl3 without Mettl14 around nearby?*

Response: Thanks for this comment. On the one hand, several previous studies revealed that METTL3 can function independently of METTL14^{1, 2, 3}. Wang et al. claimed that both METTL3 and METTL14 exhibited m⁶A methyltransferase activity independent on each other both *in vivo* and *in vitro*, although the METTL3-METTL14 complex showed stronger methyltransferase activity. Barbieri et al. claimed that METTL3, independently of METTL14, associated with the transcription start sites of active genes. These reports suggest that METTL3, somehow, might regulate m⁶A modification on VSV RNA without METTL14 in cytoplasm. On the other hand, we also hypothesize that a specific viral protein or a certain cytosolic protein might alternatively act as a cytoplasmic partner for METTL3, which could be discussed in the future direction.

2). *Although not necessary, can the authors show some minimal data (in vitro can be sufficient) on another RNA virus beside VSV?*

Response: Thanks for this suggestion. We have followed this advice to analyze the METTL3 cytosolic translocation phenotype through immunofluorescent assay with another RNA virus-SeV, and found that SeV could also induce the cytosolic translocation of METTL3 in HeLa cells (**Rebuttal Figure 1a**). We have added this new data in the **revised Figure S4b**.

We also tested another DNA virus HBV for METTL3 localization but did not find METTL3 cytosolic translocation after HBV infection (**Rebuttal Figure 1b, Revised Figure 3d**).

Rebuttal Figure 1. Immunofluorescent results indicate the endogenous METTL3 localization upon SeV (a) or HBV (b) infections at various timepoints.

Reviewer #2

In this study, Qiu et al. propose that N6-methyladenosine RNA modification of viral dsRNA serves as a major suppressor of the innate immune sensing signalling cascade. They use a combination of transient overexpression and knock down of the m6A methylase METTL3 in a range of cell lines and find that not only is the type I IFN response suppressed by METTL3 but also innate immune sensing upstream of IFNB transcription in response to VSV infection. Using knockout of METTL3 in monocytes and hepatocytes in a mouse model, they were able to confirm these observations in vivo, demonstrating reduced viral titres and increased type I IFN response after viral challenge. Furthermore, they show that METTL3, usually found in the nucleus of cells, translocates to the cytoplasm and co-localises and interacts with viral RNA as well as m6A RNA. Concurringly, they find that knock down of METTL3 corresponds with higher levels of dsRNA and lower levels of m6A RNA in infected cells. Finally, using PAR-CLIP and miCLIP sequencing techniques, they find that although both RIG-I and MDA5 sense and bind to VSV RNA, regions of high binding corresponded to regions with low levels of m6A modification, and regions with high levels of m6A modification corresponded to regions with poor interaction to RIG-I and MDA5. They conclude that METTL3-mediated m6A modification of viral RNA remodels viral dsRNA structures, preventing effective recognition by RLRs and suppressing the type I IFN response to viral infections. Overall, this study provides a novel insight into how detection of viral RNA structures is regulated by PRRs like RIG-I and MDA5 and how there are intrinsic negative regulators of viral sensing. However, despite the overall value of this study to the field, there are some major and minor issues with the manuscript that need to be addressed.

Response: Thanks for this thoughtful comment. We have performed a series of additional experiments to support the specificities of antibodies and the influence of METTL3-mediated m⁶A on dsRNA structural formation. The detailed evidence has been included in the following point-to-point responses.

Major issues:

1). *The manuscript fails to meet acceptable standard of English language and is in a need of professional proof reading and editing prior to resubmission. At times, unintentional wrong choice of words leads to sentences actually stating the opposite of what the authors presumably intend to say (e.g. lines 41-44: “The innate immune system is highly conserved among plants and animals, which recognizes the invading pathogens through pattern recognition receptors (PRRs) to trigger an effective immune response for defending the pathogens.”).*

Response: We apologize for these typos and grammatical errors. We have checked and revised the manuscript carefully with the assistance of native English speaker from Springer Nature Author Services.

2). *For IP experiments (RIG-I/MDA5) no control protein was pulled down. It is commonly known that many proteins have random RNA-binding ability. To exclude that these specific RNA-binding profiles of viral RNA are not simply random background binding, these experiments would require a negative control protein for the Ips (Figure 4f, Figure 6a).*

Response: Thanks for this comment. Actually, in the original **Figure 6e**, we included IgG as the negative control when performing the IP experiments (RIG-I/MDA5) and found that IgG could not pull down VSV RNAs. Besides, as the data showed (**Rebuttal Figure 2**), IP-western blot results also reflect the great specificities of antibodies for the IP experiment. In addition, we have precleared and blocked the beads to exclude unspecific binding before the IP assay when performing the PAR-CLIP-seq. Moreover, according to series of previous studies, it seems unnecessary to conduct PAR-CLIP-seq for the IgG group ^{4, 5, 6, 7, 8, 9}. Therefore, the existing data provides sufficient

evidence that RIG-I/MDA5/ METTL3 showed specific RNA binding ability to VSV RNA. We have added these new data in the **Revised Figure S5d (Rebuttal Figure 2a)** and **S7b (Rebuttal Figure 2b)**.

Rebuttal Figure 2. a. IP-western blot assay for Mettl3 protein by using RAW264.7 cell lysate to show the specificity of Mettl3 antibody. IgG as a negative control. **b.** IP-western blot assay for Rig-I and Mda5 proteins by using RAW264.7 cell lysate to show the specificity of Mettl3 antibody. IgG as a negative control.

3). The authors make claims that the dsRNA 'structure' is significantly altered by m⁶A modification. While this is a reasonable hypothesis, they do not offer any actual structural or biochemical data supporting this hypothesis.

Response: Thanks for this thoughtful comment. In the original **Figure 4e**, anti-dsRNA (J2) IP-qPCR results indicated that METTL3 deficiency leading to a decrease of m⁶A modification level, which results in increased VSV dsRNA level and number upon VSV infection. This suggests that METTL3-mediated m⁶A modification is important for VSV dsRNA formation. Besides, we carried out anti-dsRNA (J2)-IP and then run nuclear acid gel to detect the binding affinity between J2 antibody and synthesized biotin-labeled viral RNAs +/- m⁶A. As **Figure S7e**, m⁶A modification indeed impairs the binding ability of J2 antibody and synthesized VSV RNA oligo. Since J2 antibody, similar with RLRs, can specifically recognize dsRNA structure while not ssRNA, this result suggests that m⁶A modification impairs dsRNA formation on VSV RNA to attenuate RLRs sensing. On the other hand, our manuscript highlights that m⁶A motif functions as a cis-acting element to control innate sensing through regulating exogenous viral dsRNAs formation.

Additionally, we did IF for dsRNA in WT and METTL3 KD HeLa cells. As the result shown, shMETTL3 cell line accumulated more dsRNA than control cells upon VSV infection for 12h, which suggested that METTL3-mediated m⁶A inhibited dsRNA formation upon VSV infection. We have added these new data in the **Revised Figure 5a**.

Rebuttal Figure 3. Immunofluorescent analysis of dsRNA level by J2 antibody after PBS or VSV infection for 12h in shNC and shMETTL3 HeLa cells.

Minor issues:

1). Why did this study focus on using murine macrophages for many of the studies? In general, experiments were spread across murine macrophages, HEK-293T cells (whose suitability to study the innate immune response is dubious at best) and HeLa cells.

Response: Thanks for this question. First, innate immune response is highly conserved among human and mouse. Second, we intended to confirm that the inhibitory role of METTL3 in innate sensing is a broad and global behavior by using various cell types in this study. In addition, there were also some other studies using HEK-293T cell as cellular model for innate immunological studies ^{10,11}.

2). *Figure 1d/e/f/g: p-IRF3 Western blots are mostly not very convincing in supporting the claims made in the text.*

Response: Thanks for this thoughtful comment. These data revealed that transient overexpression of METTL3 suppressed VSV-induced IRF3 phosphorylation in different cell lines, suggesting this phenotype is conserved. Our explanation for this comment is that different cell types have different intrinsic innate immune responses and sensitivities for infections, therefore, these p-IRF3 western blots showed different intensities but similar trends.

3). *Figure 1i: there is no control where METTL3 isn't overexpressed and where IRF3 translocation isn't blocked.*

Response: Thank you for pointing out this neglect. We have pointed out the control cells in which METTL3 isn't overexpressed and where IRF3 translocation isn't blocked by using '*' in the (**Rebuttal Figure 4, Revised Figure 1i**).

Rebuttal Figure 4. Immunofluorescence analysis of IRF3 translocation and activation in HEK293T cells after VSV infection for 6h or PBS treatment. The arrowed cells indicated the METTL3 overexpressed cells which inhibited the nuclear translocation and activation of IRF3. The "*" labeled cells indicated control cells. α -Tubulin indicated the cytosolic part.

4). *Figure 1i (graph): colours aren't defined (red / black) and it is not clear where the images of those control cells are (not shown).*

Response: Thanks for pointing out this neglect. The 'red' means 'control cells', the 'black' means 'OE-METTL3 cells'. We have revised this panel (**Rebuttal Figure 5, Revised Figure 1i**).

Rebuttal Figure 5. Statistics of nuclear localization of IRF3 in revised Figure 1i.

5). *Figure 1l: the text (lines 124-126) mentioned up- and downregulated genes in infected cells METTL3-depleted cells compared to mock infected cells. However the figure panel shows RPKM values for infected KO vs WT cells. An alternative way of visualizing the data might be more accurate or acceptable, for example volcano plots taking p values into account and depicting fold change over mock infected cells.*

Response: Thanks for pointing out this. We have followed the reviewer’s advice and added the extra volcano plot which is much more accurate to show the fold changes (**Rebuttal Figure 6**). We have also included these new data in the **Revised Figure S2a**.

Rebuttal Figure 6. Volcano plots of genes with differential expression in *Mettl3* knockout and WT RAW264.7 cells after VSV infection for 12 h. (yellow: up-regulated; blue: down-regulated; grey: no significant change; red triangle: ISGs).

6). *Figure 2b and 2c:* It’s not immediately apparent what we are looking at and whether this image is in any way representative of the phenotype. Is there any way to quantify these images to get a sense of how valid the authors’ claims are?

Response: Thanks for pointing out this. H&E staining could indicate the inflammatory infiltration level in the tissues. Because the characteristic of immune cells contains big nucleus (stained by hematoxylin) but less cytoplasm (stained by eosin), with the dark zones in the tissues indicating the inflammatory infiltration. We have analyzed the inflammatory infiltration and showed the quantification in the **Revised Figure 7**. We have also included these new data in the **Revised Figure S3a**.

Rebuttal Figure 7. The statistics of Figure 2b (Left) and Figure 2c (Right).

7). *Figure 2d:* The statistics on the error bars especially for Liver and Lung *Ifnb1* mRNA levels looks dubious considering the variance of datapoints.

Response: Thanks for pointing out this mistake. The statistics of the error bars should have been ‘mean with SEM’. We have revised these graphs in the revised Figure legends and manuscript.

8). Line 154: the authors claim “resistance” to VSV-induced lethality although Figure 2i shows that there was some mortality.

Response: Thanks for this comment. We have changed the word “resistant” to “decreased” in the revised manuscript.

9). Lines 169-174: 'other' treatments are mentioned, however only a DNA virus (HSV) and poly(I:C) and poly(dA:dT) treatment is shown at a single timepoint. I would hesitate to make the authors' statement that this is specific to VSV infection without testing this in other cytoplasmic and nuclear RNA viruses as well as poly(I:C)/poly(dA:dT) treatment at various timepoints. It is likely that, since different viruses and treatments have different kinetics, that by simply looking at a single time point different phenotypes are missed.

Response: Thanks for this thoughtful comment. We carried out IF for endogenous METTL3 upon different treatments in various timepoints. As the data shown, HBV, HSV and Poly(dA:dT) and Poly(I:C) treatments cannot influence METTL3 subcellular location at various timepoints (**Rebuttal Figure 8**). However, we found that another RNA virus-SeV could also induce METTL3 cytoplasmic translocation (**Rebuttal Figure 1**). We have added this new data in the **Revised Figure 3d and Figure S4b**.

Rebuttal Figure 8. Immunofluorescent results indicate the endogenous METTL3 localization upon different treatments at indicated timepoints in HeLa cells. AC12 cells were infected with HBV.

10). Lines 212-216: It is not clear to me how the finding that specifically only positive-sense viral RNA was associated with METTL3 supports their hypothesis that VSV infection triggers cytosolic translocation of METTL3 to interact with VSV RNA.

Response: Thanks for this question. Because VSV is a kind of negative-sense RNA virus, which uses its positive-sense RNAs for translation and synthesis of viral genome, and this complicated life cycles progress lead to higher structures of viral RNAs. Furthermore, most of the negative-sense (genomic) RNA of VSV could be bound by N protein to compete the inner interaction.

11). Figure 6b is not mentioned in the main text.

Response: Thanks for pointing out this neglect. We have revised this in the manuscript. (**Revised Manuscript, pages 9, lines 254-256**).

Reviewer #3

The manuscript entitled “N6-methyladenosine RNA modification suppresses antiviral innate sensing pathways via I reshaping double-stranded RNA modules” by Qiu et al. described a potential role of the m6A methyltransferase METTL3 in suppressing Type I interferon response by reducing the sensing efficacy of RIG-I and MDA5 to double strand RNA.

Overall, the work presented is high-quality, the findings are novel and most conclusions are well-supported by experimental evidence. However, the mechanisms for regulating the METTL3 activity and the specificity of METTL3 involving type I interferon responses to VSV versus other viruses or RNA versus DNA are not clear. The manuscript also needs to be edited and revised, because there are significant issues with usage and grammar that interfere with meaning and readability.

Response: We thank the editor’s approval of our work. We have performed a series of additional experiments to show the effect of METTL3 and cytosolic METTL3 in various treatments-induced type I IFNs. The detailed evidence has been included in the following point-to-point responses. In addition, we have edited and revised the manuscript with the assistance of native English speaker from Springer Nature Author Services.

1). In figure 1a, METTL3 overexpression blunts IFN β 1 response to several stimuli/infection conditions. However, the mechanisms proposed in the rest of the manuscript such as the effects of METTL3 on dsRNA formation could not explain the observation that METTL3 overexpression inhibits IFN β 1 response to poly dA:dT, LPS and DNA viruses. What effect does METTL3 overexpression have on IRF3 phosphorylation upon infection with another RNA virus? DNA virus? Stimulation with RNA or DNA analog?

Response: Thanks for this thoughtful comment. A previous study indicated that METTL3-mediated m⁶A on *IFNB1* mRNA could regulate the mRNA stability of *IFNB1* without influencing innate immune upstream (p-TBK1-p-IRF3) upon DNA virus HCMV infection¹². Our explanation is that poly I:C, poly (dA:dT), LPS could also induce the expression of *IFNB1* mRNA which could be regulated by METTL3-mediated m⁶A modification through mRNA metabolism. In our study, we also detected the m⁶A modification on *IFNB1* mRNA, but more importantly, we found that the upstream of innate immunity pathway is strongly controlled by METTL3-mediated m⁶A modification upon RNA virus VSV infection (the original Figure 1b and c). Therefore, we focused on the mechanism of how m⁶A regulates innate immune sensing upon VSV infection.

Since dual-luciferase reporter assay could greatly demonstrate the activation of transcriptional factor IRF3, we performed dual-luciferase reporter assay to detect IFN- β promoter activity upon infection with another RNA viruses (HCV, SeV), DNA viruses (HBV, HCMV, HSV) or stimulation with RNA or DNA analog. The results showed that despite METTL3 overexpression inhibited the expression of *IFNB1* mRNA under many stimuli (**the original Figure 1a**), only RNA viruses SeV and VSV inhibited the activation of IRF3 significantly (**Rebuttal Figure 9**). We have added this new data in the **Revised Figure 1b and Supplementary Figure 1b**.

Rebuttal Figure 9. a. METTL3 overexpression (Left) or knockdown (Right) verified by qPCR. **b.** IFN- β promoter activity in HEK293T cells transfected with METTL3 vector upon different treatments for 12 h (Left). IFN- β promoter activity in AC12 cells transfected with siMETTL3 upon HBV (DNA virus) treatment for 12 h (Right).

2). It would be good to see the equivalent of figures 1e-h some of these conditions, (and/or a modified supplemental figure 4b) because the rationale for focusing solely on VSV after this is unclear. Figure 3b and 3c present a clear result: endogenous METTL3 moves from the nucleus to the cytoplasm in VSV infection. These and supplemental figure 4b are referred to in the text (line 169-170) as evidence that METTL3 doesn't move to the cytoplasm in conditions other than VSV infection, but S4b uses overexpressed METTL3 rather than endogenous METTL3. That's fine for supplemental figure 1j, which shows that METTL3 overexpression stops IRF-3 nuclear translocation in VSV infection, but it's not valid evidence for comparison to 3b/c to make the VSV-specific claim. Western blots as in 3b would be best. On the other hand, the addition of an IRF3 staining (as in supplemental figure 1j) would make supplemental figure 4b much more useful and could effectively concerns with these figures.

Response: Thanks for this thoughtful comment. We repeated the **Figure S4b** IF experiment for endogenous METTL3 upon different treatments in various timepoints. As the data showed, HBV, HSV and Poly(dA:dT) and Poly(I:C) treatments cannot influence METTL3 subcellular location at various timepoints (**Rebuttal Figure 8**). (This experiment is also used to address the minor point 9 of Reviewer 2#). However, we found that another RNA virus-SeV could also induce METTL3 cytoplasmic translocation. This result is referred to the dual-luciferase reporter assay result that SeV-induced IRF3 activation could also be inhibited by METTL3 overexpression (**Rebuttal Figure 9**).

We have added the luciferase assay results to the **revised Figure 1b**. Since VSV-induced IRF3 activation could significantly inhibited by METTL3 overexpression, we focused on VSV as an RNA virus model for further investigation.

3). The model the authors present is not solid without data to show the effect of NLS mutant overexpression on IRF3 translocation. Is cytoplasmic METTL3 alone not sufficient for the loss of nuclear IRF3 in non-VSV infection conditions (or stimulation conditions) or is the presence of VSV also required?

Response: Thanks for this thoughtful comment. We performed dual-luciferase reporter assay to detect IRF3 activation after stimulated by viruses or stimuli, and found that SeV and VSV-induced IRF3 activation could be inhibited by METTL3 but not other stimulations (**Rebuttal Figure 9**).

Furthermore, the luciferase assay result for overexpression of mutated-NLS demonstrated that overexpression of NLS-mutated METTL3 was sufficient to suppress the upstream of innate immunity by VSV infection (**Rebuttal Figure 10a**) and SeV (**Rebuttal Figure 10b**). But OE-NLS-mut did not show stronger inhibitory ability than OE-METTL3. We speculated that the cytosolic METTL3 may also have other effects on IFN- β mRNA metabolism, which led to OE-NLS-mut showed more effective than OE-METTL3 in suppression of *IFN β 1* mRNA expression while not IRF3 activation. We have added this new data in the **Revised Figure 3i** (**Rebuttal Figure 10a**).

Rebuttal Figure 10. IFN- β promoter activity in HEK293T cells transfected with vectors upon VSV or SeV infection.

4). As the authors acknowledge, there isn't any direct evidence presented of a virus-originated mechanism for METTL3 translocation or dsRNA methylation in VSV infection. However, catalytic METTL3 is known to complex with METTL14, which recognizes methylation targets. Is there a viral protein acting as a cytoplasmic partner for METTL3? The manuscript would be better if the authors checked for physical interaction between METTL3 and VSV proteins, and VSV-L is particularly interesting as a possible partner given its role in VSV's atypical mRNA capping and SAM-dependent methylation. This might be discussed as a future direction.

Response: Thanks for this thoughtful suggestion. On the one hand, several previous studies revealed that METTL3 can function independently of METTL14^{1,2,3}. Wang et al. claimed that both METTL3 and METTL14 exhibited m⁶A transferase activity independent on each other, although the METTL3-METTL14 complex showed stronger methyltransferase activity. Barbieri et al. claimed that METTL3, independently of METTL14, associated with the transcriptional start sites of active genes. These reports suggest that METTL3, somehow, could regulate m⁶A modification on VSV RNA without METTL14 in cytoplasm. On the other hand, this is a very interesting hypothesis that whether VSV proteins interacted with METTL3 in the cytosol upon VSV infection. Besides, we also hypothesize that whether another host cytosolic protein could alternatively act as a cytoplasmic partner for METTL3. As this work so far is more focused on the innate immune sensing, we will consider this suggestion in the future direction. Thanks again!

A few minor points:

1). On lines 102-108, the explanation of how these results differ from the previous report of METTL3 impact on IFN β could be more clear.

Response: Thanks for this thoughtful suggestion. We have revised this sentence as "Consistent with previous reports, we also observed an enhanced downstream of innate immunity, phosphorylation of ISGs-regulated protein Stat1, in *Mettl3*-deficient macrophages upon VSV infection, which was supposed to be increased by a decrease of m⁶A modified *Ifnb1* mRNA. However, more importantly, the phosphorylation of upstream kinase *Tbk1*, transcription factors *p65* and *Irf3*, was also significantly increased in *Mettl3*-deficient macrophages, indicating a global influence for innate immune signaling cascades by *Mettl3*." (**Revised Manuscript, pages 4, lines 101-108**).

2). Given the adverse effect of m⁶A modification on translation rate (Slobodin et al. in *Cell*, 2017), is there a tradeoff involved in this apparent viral strategy?

Response: Thanks for this valuable question. First, we agree with the reviewer's opinion that there may be a tradeoff involved in this apparent viral strategy. Besides, Slobodin et al. claimed that methyltransferases-mediated m⁶A modification on the transcriptional RNAs (maybe nascent RNAs) from chromatin in nucleus impeded mRNA translation rate¹³. Differently, the synthesis of VSV positive-sense RNAs only occur in the cytosol, therefore, m⁶A modification in different space-times may have different functions in RNA metabolism including translation process. On the other hand, the aim of viral infection is not to kill host cells but to replicate using host energy. Overuse of host energy may kill host cells in a short time, which is not an efficient and smart way for viral replication. Thus, to keep a balance, virus may use m⁶A modification to slow down viral RNAs translation rate and to escape host innate immune sensing.

3). As another point of discussion, do the author's findings regarding the m⁶A modification-induced blunting of antiviral response indicate any application potential (ie mRNA delivery into cells?)

Response: Thanks for this thoughtful suggestion. In the future direction, we may target cytoplasmic METTL3 or

inhibit METTL3 activity to ameliorate anti-viral innate immune response and cure patients who suffered from viral infections. Besides, innate immunity activation is important for the initiation of anti-tumor immunity, therefore, it is also a potential strategy for targeting METTL3 to enhance the immunogenicity of solid tumors and promote innate immunity-induced T cell infiltration in tumors. We have revised the discussion in the revised manuscript (**Revised Manuscript, pages 12, lines 364-369**).

4). The term “respectively” is used unnecessarily several times. It’s used correctly on lines 49 and 559.

Response: We apologize for these typos and grammatical errors. We have checked and revised the manuscript carefully.

5). In Figure 7, it looks like “less structures” should be “loss of structure.” “Keep structures” might be better as “intact structure.”

Response: Thanks for this thoughtful suggestion. We have revised our working model as below (**Rebuttal Figure 11**).

Rebuttal Figure 11. In the proposed model, VSV transcripts contain dsRNA structures to initiate innate immunity sensing. During the VSV infection, METTL3 can be attracted from nucleus to cytoplasm to contact and modify VSV RNA. This m⁶A modification impairs the conformation of duplex structures in VSV RNA and interferes the sensing by dsRNA sensors involving RIG-I and MDA5, which attenuates innate immune response and helps virus invasion. When the host deficient for METTL3, there are more dsRNA structures recognizing by RLRs to drive the expression of type I IFNs, following enhances anti-viral function.

References:

1. Barbieri I, *et al.* Promoter-bound METTL3 maintains myeloid leukaemia by m(6)A-dependent translation control. *Nature* **552**, 126-131 (2017).
2. Weng H, *et al.* METTL14 Inhibits Hematopoietic Stem/Progenitor Differentiation and Promotes Leukemogenesis via mRNA m(6)A Modification. *Cell Stem Cell* **22**, 191-205 e199 (2018).
3. Wang Y, Li Y, Toth JI, Petroski MD, Zhang Z, Zhao JC. N6-methyladenosine modification destabilizes developmental regulators in embryonic stem cells. *Nat Cell Biol* **16**, 191-198 (2014).
4. Zhang C, *et al.* m(6)A modulates haematopoietic stem and progenitor cell specification. *Nature* **549**, 273-276 (2017).
5. Xiao W, *et al.* Nuclear m(6)A Reader YTHDC1 Regulates mRNA Splicing. *Mol Cell* **61**, 507-519 (2016).
6. Li A, *et al.* Cytoplasmic m(6)A reader YTHDF3 promotes mRNA translation. *Cell Res* **27**, 444-447 (2017).
7. Shi H, *et al.* YTHDF3 facilitates translation and decay of N(6)-methyladenosine-modified RNA. *Cell Res* **27**, 315-328 (2017).
8. Yoon JH, *et al.* PAR-CLIP analysis uncovers AUF1 impact on target RNA fate and genome integrity. *Nat Commun* **5**, 5248 (2014).
9. Hafner M, *et al.* Transcriptome-wide identification of RNA-binding protein and microRNA target sites by PAR-CLIP. *Cell* **141**, 129-141 (2010).
10. Wang S, *et al.* YAP antagonizes innate antiviral immunity and is targeted for lysosomal degradation through IKKvarepsilon-mediated phosphorylation. *Nat Immunol* **18**, 733-743 (2017).
11. Zhang Q, *et al.* Hippo signalling governs cytosolic nucleic acid sensing through YAP/TAZ-mediated TBK1 blockade. *Nat Cell Biol* **19**, 362-374 (2017).
12. Roni Winkler EG, Lior Lasman, Modi Safra, Shay Geula, Clara Soyris, Aharon Nachshon, Julie Tai-Schmiedel NF, Vu Thuy Khanh Le-Trilling, Mirko Trilling, Michal Mandelboim JHH, Schraga Schwartz and Noam Stern-Ginossar. m6A modification controls the innate immune response to infection by targeting type I interferons. *Nature immunology* **20**, 173-182 (2019).
13. Slobodin B, *et al.* Transcription Impacts the Efficiency of mRNA Translation via Co-transcriptional N6-adenosine Methylation. *Cell* **169**, 326-337 e312 (2017).

REVIEWERS' COMMENTS

Reviewer #1 (Remarks to the Author):

The authors have addressed my concerns and comments.

Reviewer #3 (Remarks to the Author):

The revised manuscript has addressed most of the issues raised during my initial review.